# Propensity of selecting mutant parasites for the antimalarial drug cabamiquine

Eva Stadler [1,11], Mohamed Maiga [2,11], Lukas Friedrich[3,11], Vandana Thathy[4,5,11], Claudia Demarta-Gatsi [6], Antoine Dara [2], Fanta Sogore[2], Josefine Striepen [4,10], Claude Oeuvray[6], Abdoulaye A. Djimdé [2], Marcus C. S. Lee [7,8], Laurent Dembélé[2] ✉, David A. Fidock [4,5,9] ✉, David S. Khoury [1] ✉ & Thomas Spangenberg [6] ✉

We report an analysis of the propensity of the antimalarial agent cabamiquine, a *Plasmodium*-specific eukaryotic elongation factor 2 inhibitor, to select for resistant *Plasmodium falciparum* parasites. Through in vitro studies of laboratory strains and clinical isolates, a humanized mouse model, and volunteer infection studies, we identified resistance-associated mutations at 11 amino acid positions. Of these, six (55%) were present in more than one infection model, indicating translatability across models. Mathematical modelling suggested that resistant mutants were likely pre-existent at the time of drug exposure across studies. Here, we estimated a wide range of frequencies of resistant mutants across the different infection models, much of which can be attributed to stochastic differences resulting from experimental design choices. Structural modelling implicates binding of cabamiquine to a shallow mRNA binding site adjacent to two of the most frequently identified resistance mutations.

*Plasmodium falciparum* drug resistance poses a constant threat to effective malaria treatment. Consequently, resistance is a critical parameter to monitor during anti-infective drug discovery and development, as it can lead to the demise of first-line treatments[1–3]. To mitigate the risk of resistance, new antimalarial drugs should be developed as fixed-dose combinations. A better understanding of resistance risks posed by a candidate compound under development, in addition to how this will translate in a real-world setting, can inform the selection of partner drugs and the optimal design of combination therapies.

Over the last decade, the antimalarial drug research community has developed robust tools to assess antimalarial drug candidates in pre-clinical and early phase clinical studies. These include the following: (i) in vitro selection of resistance by subjecting antimalarial drugs to various inocula of *P. falciparum* parasites ($10^5$–$10^9$) to determine the minimum inoculum for resistance (MIR), regularly complemented by whole-genome sequencing of mutant parasites and fitness cost studies[4]; (ii) in vivo assessment of parasite recrudescence and the presence of resistant mutants following drug treatment in a chimeric humanized mouse model (NOD/SCID/IL2rγnull [NSG]) engrafted with

[1]The Kirby Institute, UNSW Sydney, Kensington, NSW 2052, Australia. [2]Université des Sciences, des Techniques et des Technologies de Bamako (USTTB), Faculté de Pharmacie, Malaria Research and Training Center (MRTC), Point G, PB1805 Bamako, Mali. [3]Medicinal Chemistry & Drug Design Global Research & Development, Discovery Technologies, Merck Healthcare, 64293 Darmstadt, Germany. [4]Department of Microbiology and Immunology, Columbia University Irving Medical Center, New York, NY 10032, USA. [5]Center for Malaria Therapeutics and Antimicrobial Resistance, Columbia University Irving Medical Center, New York, NY 10032, USA. [6]Global Health Institute of Merck, Ares Trading S.A., (an affiliate of Merck KGaA, Darmstadt, Germany), 1262 Eysins, Switzerland. [7]Wellcome Sanger Institute, Wellcome Genome Campus, CB10 1SA Hinxton, UK. [8]Biological Chemistry and Drug Discovery, School of Life Sciences, University of Dundee, DD1 4HN Scotland, UK. [9]Division of Infectious Diseases, Department of Medicine, Columbia University Irving Medical Center, New York, NY 10032, USA. [10]Present address: Weill Cornell Medical College, New York, NY 10021, USA. [11]These authors contributed equally: Eva Stadler, Mohamed Maiga, Lukas Friedrich, Vandana Thathy. ✉e-mail: laurent@icermali.org; df2260@cumc.columbia.edu; dkhoury@kirby.unsw.edu.au; thomas.spangenberg@merckgroup.com

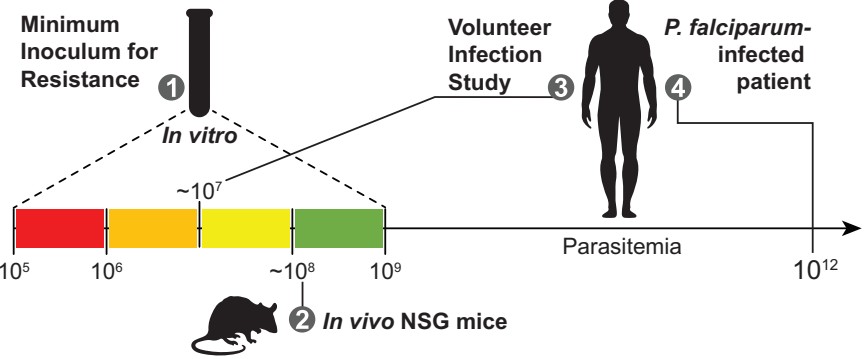

**Fig. 1 | Numbers of parasites across pre-clinical and clinical assays.** (1) In vitro *P. falciparum* minimum inoculum for resistance; (2) *P. falciparum*-infected human red blood cells engrafted in NSG mice; (3) Volunteer infection study with humans infected with blood-stage *P. falciparum* parasites; and (4) Field settings, i.e. endemic regions. NSG, NOD/SCID/IL2rγnull.

human red blood cells (RBCs) harbouring *P. falciparum* parasites (~1.4 × $10^8$ parasites/mouse at the time of treatment); and (iii) malaria naive human volunteer infection studies (VIS) with *P. falciparum* parasites (parasite loads of ~$10^7$ parasites/volunteer at treatment) to assess drug efficacy at an early stage of drug development[5]. Overall, these assays can mimic some aspects of clinical settings wherein the number of parasites can attain up to $10^{12}$ in an infected human host (Fig. 1)[6].

How well these tools enable the prediction of treatment failure due to resistance in clinical settings has been less understood. Here, we analyze pre-clinical and Phase I clinical trial data to explore the potential risks of resistance associated with cabamiquine (M5717, DDD107498)[7], an exquisitely potent inhibitor of *P. falciparum* eukaryotic elongation factor 2 (*Pf*eEF2)[8]. Mathematical modelling of these data using both deterministic and stochastic models allowed us to estimate the frequency of parasite resistance to this compound across different in vitro and in vivo infection models. This modelling also predicted the likelihood that these mutants either emerged de novo following drug treatment or alternatively were likely present at the time of treatment (pre-existent). Finally, a homology model of the target is constructed to account for the known mutant parasites and provide further insights into the binding mode of cabamiquine.

## Results

### Overlap of mutant *P. falciparum* eEF2 amino acid positions across models

To date, selection of mutant parasites with cabamiquine has been assessed in vitro, in a humanized mouse model of *P. falciparum* infection (NSG), and in human VIS[7, 9, 10]. However, to the best of our knowledge, no comparison has been performed between these infection models, either for this compound or any other antimalarial. Herein, we have identified resistance-associated mutations at 11 different *Pf*eEF2 amino acid positions (Fig. 2a and Supplementary Table 1): 8 in vitro, 6 in NSG mice, and 4 in human VIS, with 6/11 (55%) overlapping in at least two infection models. Various mutations at amino acid residues 183 and 754 were shared across all three settings (in vitro selections, and in vivo using the NSG mouse model and human VIS). The high- and medium-grade resistant mutants Y186N and P754S, respectively, were selected in the in vitro MIR studies and the NSG mouse model, whereas I183M, a low-grade resistant mutant, was only selected in vivo, both in the mouse and human hosts (Fig. 2b). The medium- and low-grade resistant mutants P754A and S474R, respectively, were recovered both in vitro and in the VIS.

Within the same *Pf*eEF2 amino acid position, different grades of resistance could be observed. For example, position 134 yielded mutant parasites in vitro with EC50 (the concentration that yields half maximal parasite growth inhibition) values ranging from 6–191 nM

depending on the mutant residue. Conversely, position 754 yielded medium-grade resistant mutants and position 186 yielded high-grade resistant mutants (Fig. 2b).

To yield *Pf*eEF2 mutant parasites, drug pressure was applied at concentrations ranging from 5× EC50 up to 297× EC50. Typically, 5× to 15× EC50 was applied in vitro (Fig. 2b and Supplementary Tables 1 and 2), whereas in the NSG mice, an estimated 276× EC50 was obtained when treating with cabamiquine at a single oral dose of 12 mg/kg[9] (Supplementary Table 3). In the human VIS, several doses (i.e. 150, 400 and 800 mg) were administered to 22 subjects, translating to an estimated 39× to 590× EC50 based on the protein binding-corrected concentration averaged over 24 h ($C_{av0-24h}$) (Supplementary Table 4). A dose of 150 mg ($n = 6$) corresponding to a mean $C_{av0-24h}$ of 53 nM led to 50% (3/6) parasite recrudescence, with one recrudescent parasite line being wild type (WT) for *Pf*eEF2 and two others having mutations in *Pf*eEF2 (at positions 474 or 183) (Supplementary Table 4). The 400 mg cohort ($n = 8$) yielded a mean $C_{av0-24h}$ of 217 nM, with 25% (2/8) of subjects having recrudescing mutant parasites (at positions 134 or 754) while the 800 mg cohort ($n = 8$) had no recrudescing parasites, with a mean $C_{av0-24h}$ of 341 nM (Supplementary Table 4).

On comparing a similar $C_{av0-24h}$ range between the NSG mouse model and the VIS studies, we observed that out of 6 subjects encompassed within a concentration range of 199–310 nM, only 1 subject ($C_{av0-24h} = 268$ nM) had recrudescent parasites. Considering all four instances of recrudescent parasites, the mutant predicted to carry the highest grade of resistance (P754A) had the highest $C_{av}$/WT EC50 ratio (297-fold). Conversely, the subject that yielded the low-grade mutant S474R had a low $C_{av}$/WT EC50 ratio (40-fold), suggesting that the higher selective pressure was associated with survival of a more highly resistant parasite.

### Translation of cabamiquine drug resistance from laboratory parasites to clinical isolates

Ex vivo susceptibility to cabamiquine was tested using culture-adapted *P. falciparum* clinical isolates obtained from 52 different Malian donors in parallel with the reference laboratory strain 3D7 (Fig. 3). Once 5–6% parasitemia was reached, parasites were continuously exposed to 7.5 nM (15× EC50) cabamiquine for 6 days. Parasitemia decreased below the limit of detection on day 4 post-drug exposure (Fig. 3a). No parasite recrudescence was observed until the end of culturing (day 50) for 50 out of 52 clinical isolates and 49 out of 52 control 3D7 cultures. These results indicate no significant difference in the propensity of field isolates to produce resistant parasites compared with 3D7 maintained in identical culture conditions (P value = 0.65). Parasite susceptibility to cabamiquine was assessed by determining the EC50 value pre- and post-drug exposure (Fig. 3b), with dihydroartemisinin used as a control (Supplementary Table 5). Comparison of cabamiquine EC50

**a**

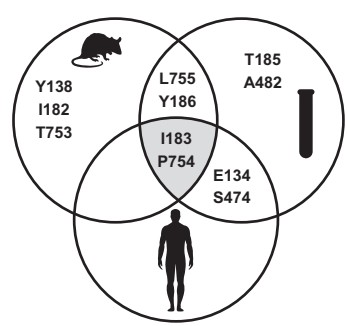

**b**

| Molecular marker | *P. falciparum* strain(s) | Setting(s) | Drug pressure (fold EC$_{50}$ of WT parasite) | EC$_{50}$ on mutant parasite |
|---|---|---|---|---|
| E134D | Dd2 | MIR | 5×EC$_{50}$ | 6 nM$^§$ |
| E134G | 3D7 | MIR | 5×EC$_{50}$ | 41–59 nM$^§$ |
| E134Q | 3D7 | VIS | 147×EC$_{50}$*** | ND* |
| E134V | Field; 3D7 | Field** | 15×EC$_{50}$ | 192 nM; 109 nM |
| Y138C | 3D7 | Mouse | 276×EC$_{50}$**** | ND* |
| I182T | 3D7 | Mouse | 276×EC$_{50}$**** | 137 nM |
| I183M | 3D7 | Mouse; VIS | 276×EC$_{50}$****; 59×EC$_{50}$*** | 2 nM; ND* |
| I183T | 3D7 | MIR | 5×EC$_{50}$ | 5 nM |
| T185I | 7G8 | MIR | 6×EC$_{50}$ | 7 nM$^§$ |
| Y186C | Field | Field** | 15×EC$_{50}$ | 2626 nM |
| Y186N | 3D7; Dd2 | Mouse; MIR | 276×EC$_{50}$****; 5×EC$_{50}$ | ND*; 3100 nM |
| S474R | 3D7; 7G8 | VIS; MIR | 40×EC$_{50}$***; 6×EC$_{50}$ | ND*; 7 nM$^§$ and 1 nM |
| A482T | Dd2 | MIR | 5×EC$_{50}$ | 7 nM |
| T753N | 3D7 | Mouse | 276×EC$_{50}$**** | ND* |
| P754A | 3D7 | VIS; MIR | 297×EC$_{50}$***; 5×EC$_{50}$ | ND*; 41 nM$^§$ |
| P754L | 3D7 | Mouse | 276×EC$_{50}$**** | 66 nM$^§$ |
| P754S | 3D7 | Mouse; MIR | 276×EC$_{50}$****; 5×EC$_{50}$ | 56 nM; 59 nM$^§$ |
| L755F | 3D7 Dd2 | Mouse; MIR; Field** MIR | 276×EC$_{50}$****; 5×EC$_{50}$ 5×EC$_{50}$ | ND*; 6 nM$^§$; 78 nM 660 nM |
| L755S | 3D7 | Mouse | 276×EC$_{50}$**** | ND* |

**Fig. 2 | *Pf*eEF2 mutations mediating varying levels of cabamiquine resistance across infection models. a** Venn diagram of *Pf*eEF2 amino acids and their locations subject to mutations under cabamiquine exposure in vitro, in NSG mouse, and in *P. falciparum* VIS. **b** Table with biomarkers of mutants selected in vitro in MIR studies (3D7, 7G8, and Dd2)[7], in culture-adapted field isolates, in *P. falciparum*-infected NSG mice (3D7)[9], or in *P. falciparum* VIS (3D7)[10]. Wild-type EC$_{50}$ values were 0.28 nM (3D7), 0.47 nM (*Pf*3D7$^{0087/N9}$ in serum), 0.24 nM (7G8), 0.19 nM (Dd2), 0.57 nM (3D7_MM), 0.46 nM (3D7_FS), 0.64 nM (EEF192), and 0.41 nM (EEF209). Low: EC$_{50}$ = 1–10 nM (blue); Medium: EC$_{50}$ > 10–100 nM (orange); High: EC$_{50}$ > 100 nM (red). *EC$_{50}$ not determined; ** field isolates (Mali, 2021); *** obtained from a free concentration ratio of the human C$_{av0–24h}$ (corrected for human plasma protein binding of 83%) and *P. falciparum* 3D7 EC$_{50}$ (corrected for Albumax binding of 45.3%); **** obtained from a free concentration ratio of the C$_{av0–24h}$ (249 nM) in NSG mice treated with a single dose of cabamiquine at 12 mg/kg (p.o.) and *P. falciparum* 3D7 EC$_{50}$. $^§$ Mixed population. MIR minimum inoculum for resistance, VIS volunteer infection studies, NSG NOD/SCID/IL2rγnull, ND not defined, WT wild type.

pre- and post-drug exposure in the *P. falciparum* 3D7 WT lines revealed 191- and 170-fold increases for 3D7_MM and 3D7_FS mutant parasites, respectively. For the untreated and treated clinical field isolates, we observed 6405- and 300-fold EC$_{50}$ shifts for the EEF209 and EEF192 mutant parasite isolates, respectively. Thus, all recrudescent parasites appeared to be less susceptible to cabamiquine. No significant differences in parasite susceptibility to dihydroartemisinin were observed for all the recrudescent parasite lines compared with the untreated parental strain (Supplementary Table 5). Based on these observations, recrudescent parasites were sequenced to identify possible *Pf*eEF2 mutations responsible for their reduced susceptibility to cabamiquine (Fig. 3c). Sequencing of the clinical field isolate EEF209, which exhibited the largest shift in EC$_{50}$, revealed a Y186C mutation in *Pf*eEF2. This amino acid change is similar to the previously reported Y186N mutation in vitro[7] and in vivo[9] in NSG mice. The laboratory strain 3D7_FS possessed a L755F mutation previously associated with medium-grade resistance[7]. Sequencing of the *Pf*eEF2 gene in the recrudescent 3D7_MM and EEF192 lines revealed an E134V mutation, which was previously reported to confer medium-grade resistance to cabamiquine. *Pf*eEF2 mutations found at the nucleotide positions 401, 557, and 2265 are displayed on the electropherogram traces for 3D7 and the field isolates EEF209 and EEF192 in Fig. 3c.

## Variable frequencies of resistant mutants between infection models

We next considered the frequency of resistant mutants present in each infection model (in vitro with 3D7 or Dd2 parasites or field isolates, NSG mouse, and human VIS), assuming that resistant mutants were present at the time of treatment. We estimated the frequency of these mutants at the time of treatment in vitro (with laboratory strains and clinical isolates), and in vivo (in NSG mouse or human VIS infection models) based on the number of parasites required to yield a resistant

mutant (see Supplementary Materials and Methods). We counted the number of instances that resistant mutants emerged across cultures/mice/individuals, given the total number of parasites at the time of treatment (see Supplementary Materials and Methods; Supplementary Tables 6 and 7). Considering the differences in infection models, parasite lines, and host cells (i.e., in the in vitro regrowth assay, donor RBCs obtained from an endemic setting were used), comparisons between estimates from the systems must be interpreted with caution. However, some comparisons were possible, and based on the in vitro MIR data, we estimated a frequency of 1 resistant mutant per $4.75 × 10^7$ parasites (95% confidence interval [CI]: $1.64 × 10^7$ to $1.38 × 10^8$) for the 3D7 strain and per $7.54 × 10^6$ parasites (95% CI: $4.42 × 10^6$ to $1.28 × 10^7$) for the Dd2 strain. The estimated frequencies differed significantly between the 3D7 and Dd2 MIR data (*P* value = 0.0007, Fig. 4a). The estimated frequency of resistant mutants in the in vitro regrowth data with 3D7 and field parasites was 1 resistant mutant per $2.42 × 10^9$ (95% CI: $7.80 × 10^8$ to $7.50 × 10^9$) and $3.80 × 10^9$ (95% CI: $9.50 × 10^8$ to $1.52 × 10^{10}$) parasites, respectively, with no significant difference between these estimates (*P* value = 0.62, Fig. 4a). In the in vivo NSG mouse and VIS study, we estimated a frequency of 1 resistant mutant per $1.20 × 10^8$ (95% CI: $5.74 × 10^7$ to $2.50 × 10^8$) and $3.67 × 10^8$ (95% CI: $1.36 × 10^8$ to $9.86 × 10^8$) parasites, respectively (Fig. 4a, Supplementary Table 6). Furthermore, no significant difference was observed between the estimates from the NSG mouse and the VIS data (*P* value = 0.065). Notably, there was a tendency for higher frequencies of resistance to be observed in the in vitro MIR studies compared with those estimated in both the field-based in vitro regrowth and NSG mouse and VIS in vivo systems. The lowest frequencies were observed in the in vitro regrowth assay, wherein host RBCs were obtained from donors from a malaria-endemic setting (Fig. 4a).

We then explored whether the differences in resistance mutant frequencies across the different experimental systems (Fig. 4) could be

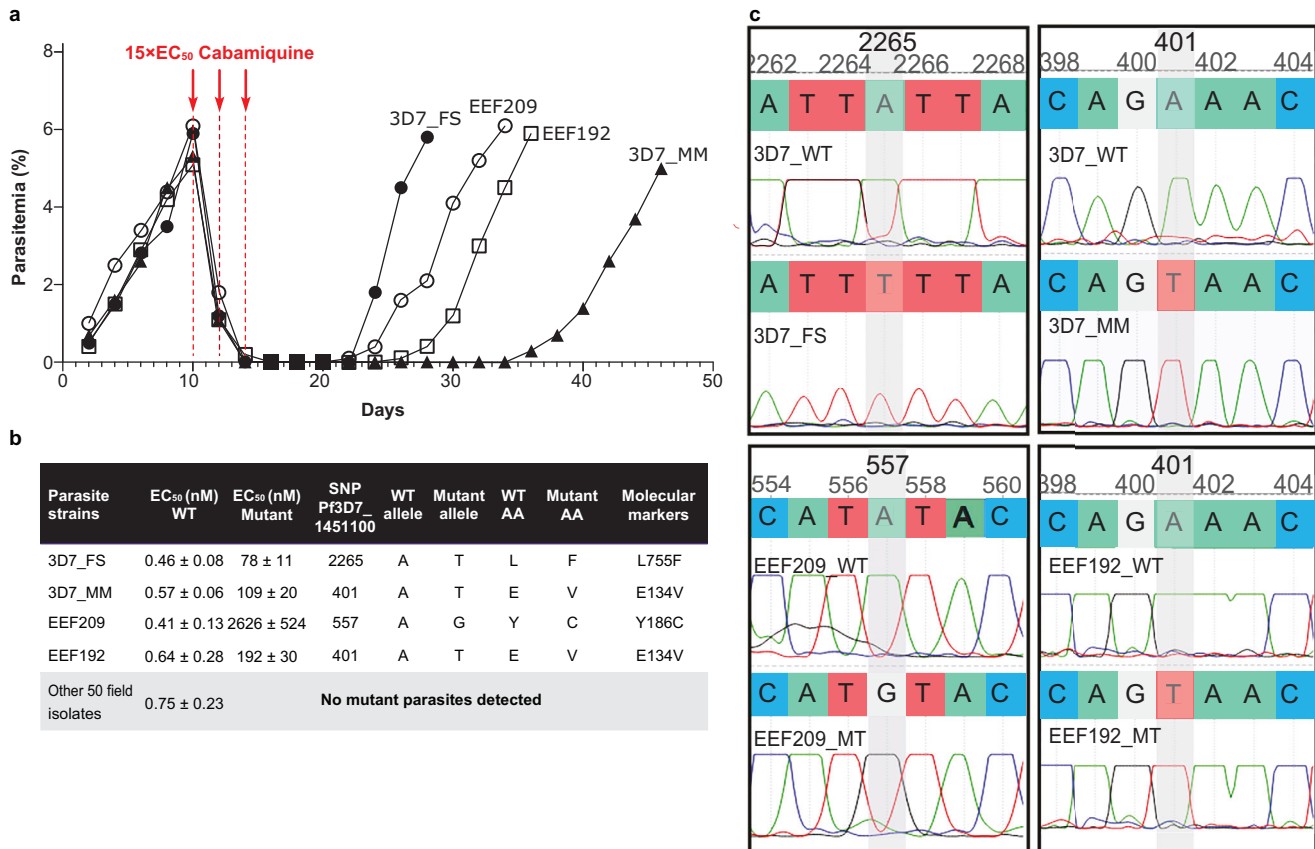

**Fig. 3 | Characterization of *P. falciparum* culture-adapted field isolates under cabamiquine drug pressure. a** In vitro *P. falciparum* asexual blood-stage parasite growth of two recently culture-adapted field isolates (EEF192 and EEF209) and two laboratory lines of 3D7 parasites (3D7_FS and 3D7_MM). Parasites were cultured for over 50 days with cabamiquine drug pressure (15× EC$_{50}$) applied for three consecutive intra-erythrocytic developmental cycles starting on day 10. **b** Table summarizing the EC$_{50}$ values pre- and post cabamiquine drug pressure for 52 field isolates, cultured in parallel with 52 sets of laboratory 3D7 parasites, along with sequencing data. **c** Electropherograms of the recrudescent field isolates (EEF192 and EEF209) and 3D7 strains (3D7_FS and 3D7_MM). WT wild-type, AA amino acid. Source data are provided as a Source Data file.

attributed to stochastic differences. For example, these differences could emerge because parasites passed through more replicative cycles in human VIS prior to treatment compared with MIR experiments. To test this hypothesis, we created a model of stochastic random mutations to determine whether the above-estimated frequencies of pre-existent resistant mutants (Fig. 4b) were consistent with the background mutation rate of *P. falciparum*. This model assumed that (1) random mutations occurred at a rate of $1.05 \times 10^{-9}$ per base pair per generation (median of four studies, Supplementary Table 10), (2) there were 11 residues in *Pf*eEF2 that when mutated could confer resistance to cabamiquine (Supplementary Table 1), and (3) resistant mutants experienced a 7% fitness cost (Supplementary Fig. 2) compared with sensitive parasites (consistent with a fitness cost per generation of 3–11% for E134D, L755F, and Y186N; see the Methods for details on the model). We used this stochastic model to simulate 100,000 in silico repeats for each experimental setting. In each simulation, we estimated the fraction of resistant parasites for each simulated experiment in the same way that we analyzed the above experimental data (see Supplementary Materials and Methods). The simulated estimates for the frequency of resistant parasites (Fig. 4a, squares and dashed lines) were highly consistent with in vitro MIR, NSG mouse, and VIS data. This suggests that the differences observed in the frequency of resistant parasites between these different infection models (Fig. 4) can be attributed to differences in experimental design rather than inherent biological characteristics that distinguish these three infection models. The exception was the in vitro regrowth studies, wherein the simulations overestimated the presence of

resistant parasites observed in vitro. This was also the only assay that used RBCs obtained from donors from a malaria-endemic setting. Thus, the overestimation by the model may be due to the simplicity of the model that, for example, does not consider differences in fitness costs across different host cell types or unmeasured fitness costs associated with the presence of innate immunity (i.e. naive volunteers versus patients from malaria-endemic countries). These factors may explain why the model estimates agree better with the estimates of in vitro MIR studies and an immunodeficient in vivo system such as the NSG mouse model.

**Resistant mutants were likely present at the time of treatment**

Importantly, in the above data analysis, we assumed that a small subpopulation of resistant parasites was present at the time of treatment (pre-existent), and that no resistance emerged de novo after treatment. Here, we extended our modelling to test the validity of this assumption. We used a deterministic formulation of the above stochastic model (Supplementary Fig. 3) to consider whether, with no preexistent resistant mutants, it was likely that resistant mutants emerged de novo only after treatment had started to inhibit parasites. For this, we assumed that when the drug was administered, viable parasite numbers reduced by a certain fraction at each replication cycle (the parasite reduction ratio, PRR) and that each generation of surviving parasites had a probability of generating resistant mutants. To be conservative, we did not limit the number of cycles of parasite replication post-treatment[11]. When a drug treatment is administered in vivo, the number of parasites reaches its peak, leading to a higher chance of

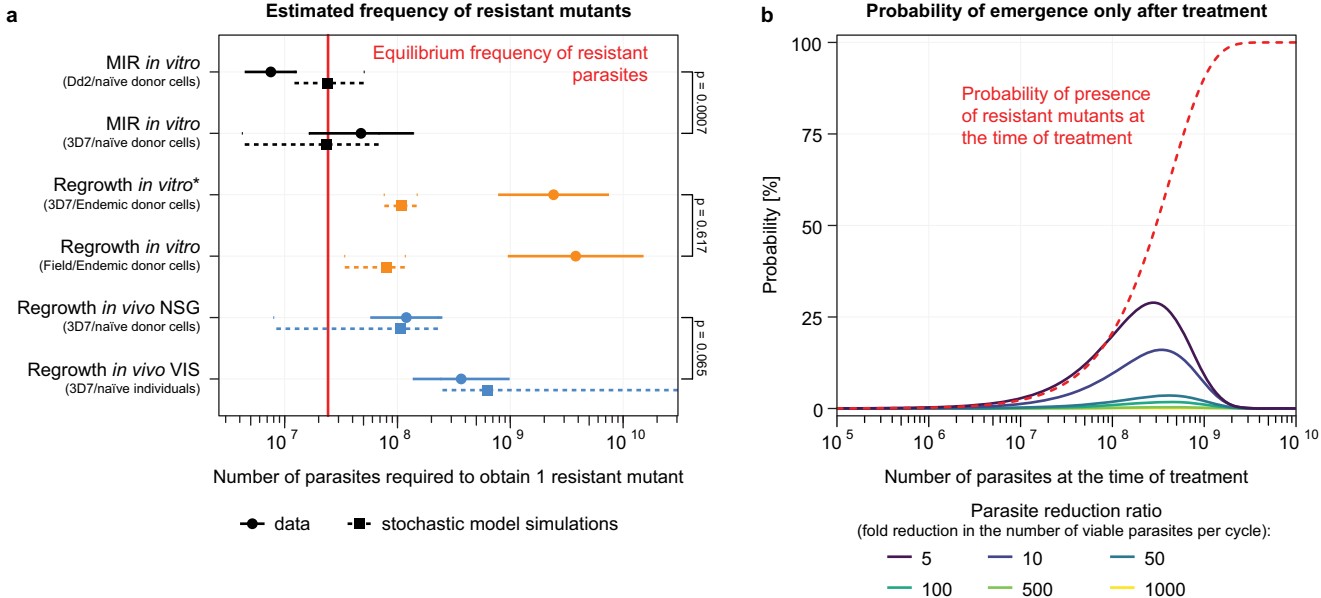

**Fig. 4 | Mathematical modelling of resistant mutants. a** Estimated frequency of resistant mutants in different infection model systems and in the stochastic model simulations of these different experimental settings. For each model system, the dot represents the estimated frequency of resistant mutants, and the horizontal line represents the 95% confidence interval. The frequency of resistant mutants was estimated using a limiting dilution assay (* for the 3D7 in vitro regrowth, we assumed that the parasite number at the time of treatment for each culture is the mean parasite number from the cultures for which the parasitemia at treatment was known, see Supplementary Material and Methods). The data differ between in vivo and in vitro experiments and between the parasite strain and RBCs used. The *P*-values on the right-hand side indicate the comparison of the respective estimates from the data using a likelihood ratio test. For the model estimates, we simulated the stochastic model 100,000 times for each experiment-specific setting (i.e. using the inoculation size, pre-treatment parasite multiplication rate (PMR), and time from inoculation to treatment for each experiment; Supplementary Table 8). For each simulated experiment, we then estimated the frequency of resistant parasites in the same way as for the data. Supplementary Table 9 lists the median (square) and the 2.5th and 97.5th percentiles (dashed line) of the estimated frequency of resistant parasites from 100,000 simulated experiments. We assumed a fitness cost of a 7% reduction in the PMR and a PMR of four per replicative cycle. We also assumed that 11 different mutations are able to mediate resistance and that the mutation rate is $1.05 \times 10^{-9}$ per base pair per generation. For estimates with different fitness costs, resistance mutation numbers, mutation rate, and PMR, see Supplementary Fig. 1. **b** The probability of the emergence of resistant mutants after treatment compared with the probability that resistant mutants were pre-existent at treatment for different parasite reduction ratios. The model predicts that the probability of resistant mutants already being present at the time of treatment is larger than that of resistant mutants emerging only after treatment for a parasite reduction ratio of ≥5. These quantities were computed using the deterministic model (Supplementary Materials and Methods). Source data are provided as a Source Data file.

resistant mutants emerging. However, parasite replication is also rapidly suppressed by the drug. Indeed, with the assumption that the mutation rate does not increase post-treatment, we found that pre-existence of resistant mutants was more likely than the emergence of resistant parasites only after treatment (Fig. 4b). Our findings suggest that pre-existent resistant mutants were likely the cause of treatment recrudescence of resistant parasites in the NSG mice and VIS hosts.

### *P. falciparum* eEF2 homology model suggests that cabamiquine binds to a mRNA binding site

Two homology models of *Pf*eEF2 were constructed using distinct methods (Supplementary Fig. 4). Through computational binding site detection, shallow grooves as potential mRNA binding sites[12] could be identified on both models close to known mutant amino acid residues (Fig. 5a, Supplementary Fig. 4E, F). Docking studies of cabamiquine revealed that poses comprising of key contacts with mutant residues could be obtained solely from the Rosetta model (Fig. 5b). The top-ranked pose and its contacts were consistent with resistance data as well as the structure–activity relationship of cabamiquine (Supplementary Table 11). The mutant residue Y186 was predicted to interact through π–π stacking with the quinoline moiety and an H-bond, and ionic bonds were predicted between the charged pyrrolidine ring and E134.

### Discussion

A thorough understanding of the propensity for cabamiquine resistance through the drug discovery and development process is key to mitigating risks of resistance emergence in field clinical settings. For example, in a Phase IIa clinical trial with the dihydroorotate dehydrogenase (DHODH) inhibitor DSM265, two *P. falciparum*-infected individuals treated with a single dose experienced parasite recrudescence at day 28 after treatment[13]. Genome sequencing of these parasites revealed a series of DHODH mutations (C276Y, C276F, or G181S). These mutations had been previously identified via in vitro resistance selections[14,15], and/or in vivo studies in which mutations were obtained in *P. falciparum* 3D7-infected NSG mice subjected to several rounds of DSM265 treatment[16]. These data suggest a good translation across experimental settings. Paramount to being able to spread and become stable within natural parasite populations, the lack of fitness defects of resistant parasite lines such as the C276Y mutant in competitive growth assays may help explain why parasites harbouring this mutation emerged readily in patients. In the VIS study, no point mutations in *Pf*DHODH could be identified, possible owing to low parasite numbers[17]. With DSM265 as an example of a slow-clearing antimalarial agent, we were curious to discover whether cabamiquine, another slow-clearing antimalarial agent, had a similar resistance profile. These data help inform the best combination partner for upcoming Phase II clinical trials.

Overall, all mutant parasites selected under cabamiquine pressure are consistent with its primary target, *Pf*eEF2, a protein that mediates the movement of the ribosome along the mRNA by promoting translocation of the transfer RNA from the A to the P site in the ribosome. Our homology model enabled us to identify a binding groove that could capture known mutants. This suggested that cabamiquine could

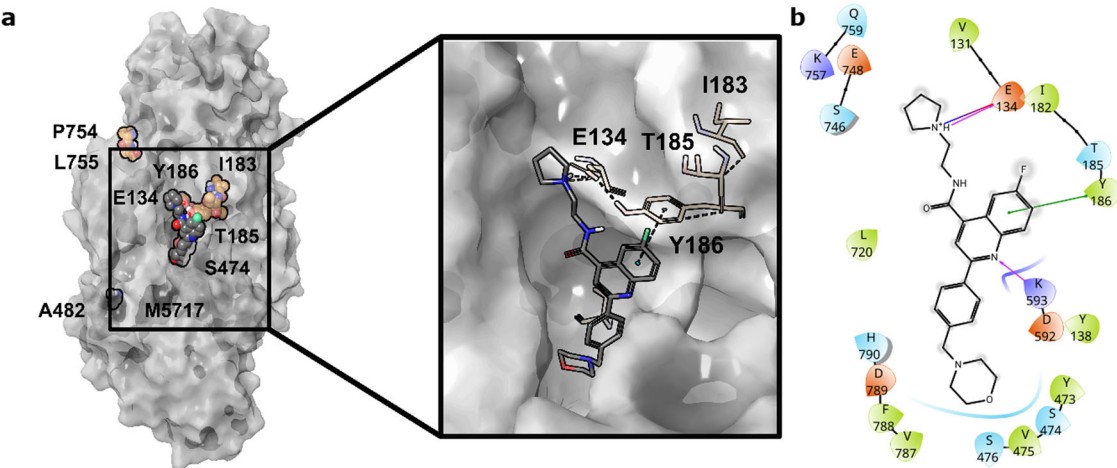

**Fig. 5 | Potential binding site and top-ranked docking pose of cabamiquine.**
**a** Surface and known mutants in the predicted protein structure of *Pf*eEF2 and potential binding of cabamiquine near residues E134 and Y186. **b** The ligand–receptor interaction map highlights amino acid residues of *Pf*eEF2 within a 5 Å radius around the potential binding pose of cabamiquine. Amino acids are coloured based on their properties (olive, hydrophobic; blue, polar; purple, positively charged; and orange, negatively charged) and interactions are shown by arrows (purple, hydrogen bonding; dark blue, salt bridge; and green, π–π stacking). Single-letter code for amino acid residues: A, Ala; C, Cys; D, Asp; E, Glu; F, Phe; G, Gly; H, His; I, Ile; K, Lys; L, Leu; M, Met; N, Asn; P, Pro; Q, Gln; R, Arg; S, Ser; T, Thr; V, Val; W, Trp; and Y, Tyr.

bind to a shallow mRNA binding site through key interactions with the amino acid residues Y186 and E134. Y186 was predicted to interact through π–π stacking with the quinoline, and mutation of the tyrosine (Y) to the cysteine (C) or asparagine (N) is consistent with the high degree of resistance as no more π–π stacking could be established. Regarding E134, this residue was predicted to interact with the charged pyrrolidine ring through an H-bond and ionic bonds. When the glutamate (E) was replaced by an aspartate (D), a one-carbon shorter homologous residue, it seems that a less optimal ion pair could still be maintained with the charged pyrrolidine moiety explaining the weak $EC_{50}$ shift (~30 fold). Conversely, upon replacement by glycine (G), the $EC_{50}$ shift was greater (~200 fold), as no interactions could occur with the charged pyrrolidine. Finally, upon replacement by a valine (V) containing a more sterically hindered isopropyl moiety, an even greater shift in $EC_{50}$ (~50–300 fold) could be achieved.

Using laboratory strains (e.g. 3D7 and Dd2), upwards of $10^7$ parasites could suffice to select for mutants resistant to cabamiquine in vitro. In addition to WT, three mutant lines derived from in vitro evolution were examined (E134D, L755F and Y186N). Each of the mutant lines showed a fitness disadvantage relative to WT (Supplementary Fig. 2A), with the fitness cost showing a trend towards correlating with the degree of resistance (increase in $EC_{50}$: Supplementary Fig. 2B). The most studied mutation, Y186N, incurs a high fitness cost (11% reduced rate of parasite proliferation per 48-h asexual blood-stage cycle), thus rendering it susceptible to be outcompeted by more fit parasites in mixed-infection settings in the absence of drug pressure. Additionally, this mutation along with P754S overlapped with the mutations obtained from the NSG mouse model study for which the inoculum load at treatment was $10^8$ parasites. Conversely, the P754A and S474R mutations overlapped with the mutations obtained in the VIS study for which the inoculum load at treatment was $10^7$ parasites. Therefore, unsurprisingly, mutants were selected and the rather high diversity of mutants across the models would indicate a stochastic mutation of the target. While the sample sizes were small in the VIS, a trend of high dose –high-grade mutant selection could be observed. Nevertheless, in a manner dependent on drug pressure, the correlation of higher-level resistance with an increased fitness cost would suggest that in a mixed infection, lower-level resistant mutants would predominate.

Our analysis highlighted that it is more likely that cabamiquine-resistant mutants were pre-existent rather than they emerged de novo after treatment at frequencies approaching 1 resistant parasite per 2.4 × $10^7$ parasites given enough time for parasites to achieve high loads. We have not explicitly considered the possibility that cabamiquine increased the mutation rate of the parasite in our analysis (i.e. that cabamiquine is mutagenic), based on the close agreement between the frequencies of resistance predicted by our stochastic model and those observed experimentally. Instead, the previously reported background rate of *P. falciparum* mutations and the number of replication cycles prior to treatment in each of these experimental systems was sufficient to explain the rates of resistance observed herein. This was consistent with previous modelling in HIV, which reported that resistance mutants were more likely to emerge from random mutation prior to strong selection pressure (such as immunity in the case of HIV) than from the same random mutation processes after the emergence of selection pressure; this can be attributed to high viral load prior to immune-mediated selection pressure[18]. Together, this result suggests that any antimalarial compound where moderate or high-grade resistance can be conferred by any one of approximately 11 single point mutations is likely to have similar frequencies of resistance to cabamiquine. This highlights the need for using cabamiquine in combination therapies as recommended by WHO and preferably in settings where low parasite burdens are expected, for example as chemoprevention.

In selecting a partner drug during the development of antimalarial combinations, there are a number of important factors to consider, such as the half-lives of each drug[19–21], drug interactions and how these will impact on overall effectiveness, and propensity of the combination to resistance[22]. In addition, one element to consider is the number of surviving parasites that would need to remain susceptible to the second drug, in the case that the partner drugs do not interact. Using the data from the 12 mg/kg dose in NSG mice, which corresponds to ~500 mg (free base) of cabamiquine used for the human dose, we estimated a frequency of about 1 resistant mutant per $10^8$ parasites. Thus, for an individual with $10^{12}$ parasites, the partner drug would contribute to reducing the number of parasites to <$10^8$ (preferably far below this level) and eliminate cabamiquine-resistant parasites to avoid recrudescence.

Based on our datasets, cabamaquine can select for pre-existent resistant parasites in naive conditions, i.e. in laboratory settings, starting from an inoculum of ~$10^7$ parasites in vitro and in vivo with *P. falciparum* laboratory strains such as 3D7 or Dd2. As the parasite strain

may influence the frequency of mutations, we evaluated an in vitro regrowth assay with 52 *P. falciparum* field isolates from Mali in parallel with 52 *P. falciparum* 3D7 selections as reference, using different RBC donors. We did not observe any differences in the number of field isolates (2/52) that developed resistance compared with the frequency of the 3D7 laboratory strain (3/52); the low numbers of resistant lines limited the power to observe a difference between these groups. Thus, we cannot exclude the possibility that propensities for resistance may differ between these field and laboratory strains. Sequencing of these mutant parasites revealed mutations in previously described resistance sites in *Pfe*EF2 (E134V, Y186C, and L755F) yielding $EC_{50}$ increases of two to three orders of magnitude. Independent of the strain (field isolates or 3D7), the selection of mutants was substantially more challenging in field-based in vitro recrudescence assays than in MIR studies. This may be due to differences in assay design between the MIR and recrudescence approaches, experimental conditions, or differences resulting from the use of RBCs from malaria-experienced or naive donors across the two assay platforms[23]. For example, variation in host cells, especially those from malaria-endemic settings, may further reduce the fitness of resistant mutants, consistent with the observations of multiple RBC polymorphisms that can reduce RBC susceptibility to *P. falciparum* infection[24–26].

A limitation of our stochastic model is that it assumes that fitness costs are a fixed quantity over time and across infection models. However, the costs of resistance are likely not fixed and may be highly dependent on the ecological conditions[27]. Although the model was relatively insensitive to the fitness cost parameter in the range tested (Supplementary Fig. 1C), fitness cost will invariably be an important factor in more extreme scenarios wherein, for example, fitness is dramatically reduced. Our current model could not consider these factors, nor could we predict with certainty that these results would translate to a field setting, as fitness costs were not assessed in the different host environments or over time. Despite these simplifications, the model could accurately predict the observed frequency of resistant mutants across different experimental systems, with the exception of the field isolate experiments. For the latter, host RBC-specific determinants and/or innate immune mechanisms may play a role in reducing the frequency of resistant mutants. Studies have demonstrated the critical interplay between the host and antimalarial drug treatment in terms of the clearance of drug-resistant parasites in patients[28,29]. We speculate that with the use of host cells from malaria-endemic residents, the fitness cost of resistant mutants may be higher. Further studies are merited to determine whether MIR values differ between field and laboratory settings.

In conclusion, cabamiquine is a *Pfe*EF2 inhibitor that appears to bind to a shallow mRNA binding site, as inferred by our homology model recapitulating the known *Pfe*EF2 mutants. Broadly speaking, we observed that the selection of mutants under cabamiquine pressure could be translated across models, suggesting that in vitro studies (MIR, field isolates, etc.) may prove useful for the assessment of resistance risks for antimalarials early in their development. Our mathematical modelling suggests that *Pfe*EF2 mutants are likely to be pre-existent. Moreover, clinical data from the VIS could indicate that high doses may select for more resistant parasites. Importantly, unlike other targets such as DHODH, it seems that the fitness cost for *Pfe*EF2 mutant parasites appears to be high, which would pose a barrier to their dissemination. Overall, this study helps understand the potential risks associated with a given drug and provides better guidance in defining selection criteria for the partner drugs to mitigate the emergence of resistance.

## Methods
### Ethics statements
The human biological samples were sourced ethically, and their research use was in accord with the terms of the informed consents.

The study was approved by the ethical committee of the Université des Sciences, des Techniques et des Technologies de Bamako (USTTB) under the reference: N° 2020/296/CE/FMOS/FAPH renewed N° 2022/03/USTTB and then N° 2023/03/USTTB.

### In vitro growth inhibitory assays of *P. falciparum* field isolates in Mali
*P. falciparum* clinical isolates from Mali and 3D7 lab strain parasites were cultured at 0.5% parasitemia/2% haematocrit in complete RPMI-1640 medium (10.43 g of RPMI-1640, 5.96 g of HEPES, 2.5 g of $NaHCO_3$, 50 mg of hypoxanthine, 5 g of Albumax, 2.5 mL of 50 mg/mL gentamicin in 1 L of $H_2O$) in 96-well plates in the presence of cabamiquine. Compounds at 10 µM were 1:3 serially diluted into eight concentration points and tested in duplicated wells. Parasites were exposed to drugs for 48 h at 37 °C under 5% $CO_2$ atmosphere. At the end of the treatment, drug susceptibility was determined by flow cytometric analysis of parasites stained with SYBR Green and Mitotracker as previously reported[30]. Fluorescence data were plotted using GraphPad Prism v.9 (GraphPad Software, San Diego, CA, USA). The data were curve fitted to a curve with a variable slope function to estimate $EC_{50}$ values. For each isolate, a Z' factor to assess assay quality was calculated from positive controls (eight drug-free wells) and negative controls (eight parasite-free, RBC control wells). Assays with a Z' values of >0.5 were considered good assays; however, each curve was visually examined for suitability. Some assays with a Z' value of <0.5 may be considered valid depending on factors such as the standard error of the curve fit $EC_{50}$. Dose–response curves and $EC_{50}$ values were calculated via non-linear regression analysis using GraphPad Prism v.9, with the data previously normalized to the untreated controls. At least three independent experiments were conducted using each compound. Wild-type samples from Mali are deposited at the ICERMALI biobank.

### In vitro evolution of cabamiquine resistance in *P. falciparum* field isolates
Mutant parasites resistant to cabamiquine were selected from Malian field isolates. A total of 52 isolates (collected in February 2021) that were adapted to culture and tested alongside the reference strain 3D7 were used for the assays. Parasites were cultured in human RBCs using complete RPMI-1640 medium[31] in 25 cm² cell culture flasks (cat# CLS430639, Corning). Cultures were maintained in a 5% $CO_2$ atmosphere at 4% haematocrit in a total volume of 6 mL in humidified modular chambers at 37 °C. When parasitemias reached ~5% with > 95% ring stages, cultures were treated with 15 × $EC_{50}$ (7.5 nM) of cabamiquine (Merck KGaA, Darmstadt, Germany) for three consecutive 48-h incubation cycles. At the end of the drug exposure, cabamiquine was removed from cultures using three consecutives washing steps with 1× phosphate buffered saline. Cultures were resuspended in complete RPMI-1640 and incubated at 37 °C under 5% $CO_2$/5% $O_2$/90% $N_2$ atmosphere. Parasite growth was monitored every 48 h using Giemsa-stained slides[32]. $O^+$ RBCs used for the parasite culture were obtained from the Malian blood bank in Bamako from adults assumed to be non-naive for prior *P. falciparum* infection. Parasite synchronization was performed by resuspending the infected RBC pellets in 10 volumes of 5% D-sorbitol (SIGMA) for 5 min[33].

### *P. falciparum* eEF2 gene sequencing
The *Pfe*EF2 gene was sequenced at the Malaria Research and Training Centre, Bamako, Mali as previously described[9]. The original protocol was modified to amplify and sequence the entire gene using the Seq-Studio Genetic Analyzer (Applied Biosystems). Briefly, genomic DNA was extracted from blood samples using the Qiagen DNA extraction kit (cat# 51306, Qiagen). The 2.5 kb *Pfe*EF2 gene was PCR-amplified using flanking primers (Supplementary Table 12). The PCR conditions for the initial amplification were as follows: 95 °C for 3 min, 45 cycles at 98 °C for 20 s, 55 °C for 30 s, and 68 °C for 2.5 min, with a final extension of

3 min at 68 °C. Agarose gel (1%) electrophoresis was used to confirm the PCR product size. The PCR product was purified using the ExoSAP-IT Express kit (cat# 78200.200.UL, Thermofisher). In addition to the amplification primers, 10 sequencing primers were used to fully sequence the PfeEF2 gene (Supplementary Table 12). The sequencing conditions for the second step were as follows: incubation at 90 °C for 1 min, following by 25 cycles of denaturation at 90 °C for 10 s, annealing at 50 °C for 5 s, and extension at 60 °C for 4 min with a ramp rate of 1 °C/s. The sequencing product was purified, resuspended in Hi-Di™ Formamide, and subjected to capillary electrophoresis using the SeqStudio™ instrument. Raw sequence data were visualized using BioEdit 7.2 /UGENE 43, trimmed, and aligned to the WT 3D7 PfeEF2 reference sequence. Mutations were determined from the alignment files and inspected using the 4Peaks V1.8 software.

## Estimation of the frequency of resistant mutants

To estimate the frequency of resistant mutants, we used four different datasets: (i) in vitro 3D7 and Dd2 MIR data (Supplementary Table 2), (ii) in vivo humanized NSG mice infected with *P. falciparum* (*Pf*3D7[0087/N9])[34–36] (Supplementary Table 3), (iii) VIS data[10] (Supplementary Table 4), and (iv) data from in vitro regrowth using 3D7 and field isolates. From the in vitro MIR data set, we used data for the 3D7 and Dd2 strains (Supplementary Table 2) and a maximum likelihood approach to estimate the fraction of resistant parasites for the 3D7 and Dd2 MIR data. We assumed that there were no resistant parasites in the inoculum if a well was negative and that there was at least one resistant parasite if the well was positive. As a sensitivity analysis, we also estimated the frequency of resistant mutants assuming that 5% of cultures/mice/individuals with resistant parasites are false negatives[37] (Supplementary Table 7). Subsequently, the likelihood of a certain fraction of resistant parasites was then computed using a binomial distribution (see Supplementary Material and Methods, Estimation of the frequency of mutants from the data, for details). For the estimated frequency of resistant mutants in the MIR data, we used the data shown in Supplementary Table 2. For the estimated frequency of resistant parasites in the regrowth data, the NSG mouse data, and the VIS data, we used the same approach as for the in vitro MIR data. We note that this method assumes independence in the probabilities of resistance among parasites from within the same well/mouse/individual (see Supplementary Material and Methods for details). The number of parasites at the time of treatment was computed assuming a blood volume of 2 mL, a haematocrit value of 70%, and a mean human RBC volume of 90 fL for the NSG mouse data and a blood volume of 5 L for the VIS data. For the regrowth data with endemic donor cells, we assumed that each culture had a volume of 6 mL with a haematocrit value of 4% and a mean human RBC volume of 90 fL. If recrudescence with resistant parasites was observed, then we assumed that there was at least one resistant parasite present at the time of treatment (see Supplementary Material and Methods, Estimation of the frequency of mutants from the data; Supplementary Table 6).

To test for differences in the estimated frequencies of resistant mutants across the different experimental conditions, we used the likelihood ratio test comparing the model with different parameters for the frequency of mutants in two datasets (different frequency of mutants) and the model with only one parameter for the frequency of mutants (same frequency of mutants). A significant *P*-value (<0.05) indicates that the model with different frequencies of mutant parasites was a significantly better model and thus that there was a significant difference between the estimated frequencies.

## Stochastic and deterministic models for the frequency of resistant mutants

We used a stochastic model to simulate the experimental settings of the different infection models. Briefly, we constructed a simple deterministic model that includes multiplication of parasites sensitive to cabamiquine, mutation to produce drug-resistant parasites, multiplication of drug-resistant parasites, and a replicative fitness cost of resistant parasites (see Supplementary Materials and Methods, Deterministic model for the frequency of resistant mutants). We then extended this model to include stochasticity in the number of progeny of each parasite and in the mutations (the stochastic and deterministic models agreed well after several generations and converged to the same equilibrium frequency of resistant parasites, Supplementary Fig. 3).

In the stochastic model, the number of progeny of each parasite was sampled from a Poisson distribution, wherein the mean was the parasite multiplication rate (PMR) of either sensitive or resistant parasites. Each emerging parasite was subject to a probability of being a drug-resistant mutant. We simulated this by sampling from a binomial distribution wherein the number of "trials" was the number of parasites that may mutate, and the "success" probability was the probability that a resistance mutation occurs. The overall number of sensitive parasites in the next generation was then the total progeny of sensitive parasites minus the number of sensitive parasites that had a resistance mutation. The overall number of resistant parasites refers to the total number of offspring of resistant parasites plus the progeny of sensitive parasites that mutated to acquire resistance.

Since the sum of independent Poisson distributions is Poisson distributed, with the mean being the sum of the means of the individual Poisson distributions, the total progeny of sensitive parasites is Poisson distributed and the mean is the sum of the PMRs (which is the same for all sensitive parasites i.e. the mean is the product of the PMR and the number of parasites that replicate).

The parameters used for both the deterministic and stochastic models varied across the different experimental conditions and a sensitivity analysis was performed. In the analysis shown in Fig. 4, a fitness cost of 7% reduction in the PMR was used (consistent with a fitness cost per generation of between 3–11% of E134D, L755F, and Y186N, Supplementary Fig. 2), along with 11 amino acid locations for resistance mutations (Supplementary Table 1), and a mutation rate of $1.05 \times 10^{-9}$ base pair substitutions per generation and base pair (median of four studies using 3D7 parasites[38–41], Supplementary Table 10). However, we also considered the effect of varying these parameter values on the frequency estimate of resistant parasites (Supplementary Fig. 1). With this stochastic model, we could also simulate the experiment-specific settings of the different infection models by using experiment-specific parameter values (see Supplementary Materials and Methods, Stochastic simulations of the different experimental settings, and Supplementary Table 8).

## Computational analysis of *Plasmodium* eEF2 homology model and binding site of cabamiquine

A 3D protein model of *Pf*eEF2 was predicted from its protein sequence (Uniprot ID Q8IKW5) using the software trRosetta (Supplementary Fig. 4A, C)[42–44]. An alternative predicted 3D model was retrieved from the AlphaFold prediction database (Supplementary Fig. 4B, D). All further calculations were performed in the Schrödinger molecular modelling suite (release version 2021-01) using default parameters unless otherwise noted. The predicted structures were prepared with the Protein Preparation Wizard using default settings, including a structure minimization with the force field OPLS3e[45] and a hydrogen-bond optimization at pH 7. Binding site predictions were performed using SiteMap[46]. Potential binding sites were detected on both predicted protein structures in proximity to known mutation sites (E134 and Y186) (Supplementary Fig. 4E, F).

Cabamiquine as a ligand was prepared using LigPrep and docked into selected pockets using Glide (mode: Single Precision)[47]. The top-ranked docking pose of cabamiquine and its interaction map are shown in Fig. 5a, b.

## Statistics & reproducibility

The number of in vitro replicates, animals or individuals in each experiment is as reported throughout the main text and summarized in the Supplementary Tables 1–3. No data was excluded from the analysis, and human studies were not randomized or blinded. Unless otherwise stated, all data fitting was performed using maximum likelihood approaches, and statistical comparison were likelihood ratio tests (a two-sided statistical test for hypothesis testing between nested models).

## Reporting summary

Further information on research design is available in the Nature Portfolio Reporting Summary linked to this article.

## Data availability

All data and code to replicate the stochastic model simulations are available on GitHub (https://github.com/estadler/cabamiquine) and Zenodo[48]. Other datasets analyzed during the current study are provided with this paper as Supplementary Material. Source data are provided with this paper.

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

## Acknowledgements

Partial funding from Merck KGaA, Darmstadt, Germany is gratefully acknowledged. D.A.F. acknowledges partial funding from Medicines for Malaria Venture (RD-08-0015), the Bill & Melinda Gates Foundation (INV—033538), and Merck KGaA, Darmstadt, Germany. We acknowledge Medicines for Malaria Venture, our collaborators at the Swiss TPH Parasite Chemotherapy Unit, and The Art of Discovery for support in generating in vivo animal data. We thank Kunal Jain, PhD (Merck Specialities Pvt. Ltd., Bengaluru, India, an affiliate of Merck KGaA, Darmstadt, Germany) for copy editing suggestions, formatting, and publication coordination assistance.

## Author contributions

T.S. conceptualized and oversaw the study and contributed to the acquisition of funding and key experimental materials. T.S., E.S., L.F., V.T., C.D.G., L.D., D.A.F., and D.S.K. designed the study. E.S., M.M., L.F., V.T., F.S., and J.S. carried out the experimental work. E.S., L.F., V.T., C.D.G., A.D., C.O., A.A.D., M.C.S.L., L.D., D.A.F., D.S.K., and T.S. performed data interpretation. T.S. and C.D.G. wrote the original manuscript. All authors contributed to the revision and editing of the manuscript.

## Competing interests

L.F. is employed by Merck Healthcare, Darmstadt, Germany. C.D.G., C.O., and T.S. are employed by Ares Trading S.A., Eysins, Switzerland, an affiliate of Merck KGaA, Darmstadt, Germany. All other authors declare no competing interest.
