## [Peer Review File · Nature Communications]

REVIEWER COMMENTS

Reviewer #1 (Remarks to the Author):

The noteworthy results here concern a potential new antimalarial drug. Resistance to it easily evolves. There are three parts to the paper. The first uses several different approaches to estimate how readily and the mutations involved in resistance. That is definitely of interest if you care about the drug, and it is of interest that different experimental systems give similar ballpark estimates of the rate, and often (but not always) different mutations. The second part (lines 207-237) attempts to estimate whether the mutations arise before or after treatment. The third part is a compelling speculation on a key binding site (lines 239-256)

So far as I can tell, there are no new data in this paper; what is new is the comparison across different experimental systems (Fig 1) and some new modeling. There is an argument that the paper is of considerable interest to malariologists and so perhaps could be in a more specialized journal. The interesting generality (beyond malaria) is really the compare and contrast of the different approaches to experimental evolution being brought to bear on the same drug (showing it is possible to attack the problem in different ways), but I am guessing the conclusions that flow from the compare and contrast are rather specific to this drug and bug (here, there is some convergence at the genetic level, but also some idiosyncrasies associated with particular experimental approaches). Nothing wrong with that specificity – for this drug, it's an important assessment for malariologists.

The paper is well written and easy to digest. There is enough detail to work out what was done etc.

The second part of the paper I find slightly odd and I would be inclined to drop that (minimally justify much better why is it important/interesting). I am not sure what it means to ask whether the resistant mutants were pre-existent or arose after drug treatment. If the null model is that the drug is not mutagenic, then I am not sure it really matters. For a fixed mutation rate and replication rate, it will boil down to how many mutations arise pre-treatment and how many arise post-treatment and that will just boil down to the population dynamics (when in the infection treatment starts, and how effective the drug is at reducing replication). It would be an interesting question to ask if the drug was mutagenic but that is not what is being done here. In the field of course the issue matters (can the drug kill all the bugs before a resistant mutant arises?), but that is not what is being calculated here.

Moreover, trying to answer that odd question requires the authors do some calculation of costs of resistance. Costs of resistance are not fixed and are highly dependent on the ecological conditions and the fitness measure involved (e.g. Huijben et al. *Evolution, Medicine and Public Health* 2018: 127-137). Even if the dynamic range of the fitness costs could be estimated properly in each experimental system,

the extrapolation to field setting would be impossible. I don't think that really matters if you buy my argument that the question itself is not very interesting. The important point is that the spontaneous mutation rate is high however you measure it, and that consequently this drug will fail unless it is combined with at least one partner drug with a similar half-life and very different mode of action.

The structural analysis (part 3, fig 6) is really interesting.

Reviewer #2 (Remarks to the Author):

This paper explores the emergence and growth of *Plasmodium falciparum* parasites exposed to the anti-malarial drug (cabamiquine) across a number of environments (in vitro, in humanized mice, in volunteer trials). The authors compare the genetics of resistant mutants identified across settings — finding some mutants are shared — and characterize their phenotypes. Using a common lab strain (3D7) and clinical isolates, in vitro, they find that most cultures are cleared under drug pressure and there is no significant difference in the propensity to evolve resistance between the lab strain and clinical isolates. Next, the authors estimate the frequency of resistant mutants at the start of treatment in the different contexts, finding pretty substantial variation (several orders of magnitude separating the lowest and highest frequencies). The authors then use a simple mathematical model to determine whether it is more likely that resistant mutants were already present at the start of treatment, or emerged afterwards, finding that the former was more likely, generally. Lastly, the authors build structural models of the parasite protein that is the drug's target, to investigate the potential impact of the identified mutations.

There is a lot happening in this manuscript and for me that obscures a little bit what the key messages are. For example, the structural part doesn't seem particularly well integrated (e.g., not really motivated in the introduction at all). The comparison of specific mutations across contexts is pretty interesting, but I'm not sure strong/novel the conclusion is that "in vitro work...may prove useful for the assessment of antimalarials early in their development". I am pretty excited about the work in between these two parts: understanding whether drug resistance comes from standing genetic variation in the parasite population at the time of treatment or de novo mutation once treatment is administered is important for tailoring treatment regimes and predicting the long-term outlook for any given drug. My main comments focus on this section, where I have some questions about the modeling and the language.

1. A large concern I have about the model for the frequency of resistant mutants is that it is a deterministic model that focuses on equilibrium fractions. Of course, when mutants first emerge in a population their growth is going to be strongly subject to demographic stochasticity. I could imagine this is one reason why the model produces a much higher frequency of resistance than most of the other scenarios. Have you considered different formulations?

2. In estimating the frequency of resistance from the data, I imagine the assumption that there were no resistant parasites in the inoculum if a well was negative (l. 408-410) could also lead to underestimating frequencies (again, due to demographic stochasticity). I assume this is standard protocol, though I wonder if the work of Helen Alexander (e.g., Alexander & MacLean PNAS 2020) might be relevant here? More broadly, I had a hard time understanding the details of these estimations.

- Is it the case that the parasites are essentially being pooled within a setting? We are estimating the frequency of resistance across, e.g., all parasites in all hosts? If I've understood that correctly, is pooling ok given that parasites within a given host would then represent not totally independent subpopulations.

- I'm not clear on the effect of the assumption in l. 418-420 (and elsewhere) that exactly one resistant parasite is present at the start of treatment. The text explains that this is necessary for computing confidence intervals, but it's not immediately obvious to me how, or how else this assumption plays into the calculations. To me, it's weird for a take home message to be that most likely resistant mutants exist at the start of treatment, but to get to that conclusion we have to assume that there can only be 1.

3. I also struggled with the evolutionary language throughout the paper. Again, I quite like the question of whether there is standing genetic variation for resistance (it is "pre-existing") or whether it emerges de novo following treatment, but the text largely avoids putting it that way.

- l. 69-70 "mutants...emerged directly as a result of drug pressure or ... as a result of background mutation rates". This sort of gives the impression that the drug may be mutagenic. Ditto for line 210 "resistance emerged as a result of drug pressure"

- l. 307 "random mutation, rather than selection pressure, could explain the presence of escape mutants". I realize I am being a pedant here, but in either case mutation is necessary!

Minor comments.

- l. 80-81 - I was a bit confused by this language because it could be interpreted as meaning the identical mutations (rather than identical residues, but different amino acids), of which none is shared across all three settings.

- Maybe not important for most reader of this paper, but I don't think EC₅₀ is defined anywhere.

- l. 212, should this be a reference to Figure 5A?

- l. 294-296, do you mean "load at treatment" rather than "inoculum at treatment"? (or perhaps simply "inoculum"?)

- l. 299-301, the phrase "in a manner dependent on drug pressure" is doing a lot of work here and "could predominate" rather than "would predominate" seems more appropriate. Having said that, could one (roughly?) quantify the conditions under which the lower-level resistant mutants would predominate? Or perhaps not if the fitness costs are determined in a no drug environment — one would need to know the relative fitness of the different resistant mutants at different drug doses?

- l. 443-444, make explicit here that another assumption is that all resistance mutations are identical phenotypically? (equal p_r?)

Reviewer #3 (Remarks to the Author):

Review of 'Propensity of Selecting Mutant Parasites for the Antimalarial Drug Cabamiquine' by Stadler et al.

Major Comments

Cabamiquine (Cab) is a promising new antimalarial drug undergoing preclinical evaluation which targets eukaryotic elongation factor 2 (EEF2) required for parasite protein synthesis. Under drug selection parasites resistant to Cab can arise and these can have mutations at 11 amino acid positions in the protein and some of these mutations are shared between parasites grown in a dish, in a mouse or a human volunteer. Modelling predicts that these mutations were likely already present in the parasite populations prior to resistance selection rather than arising after drug after selection. Structural modelling suggests these mutations cluster around EEF2's mRNA binding groove. Overall, this was an interesting paper although I feel it lacked clarity in places and could have been better presented.

Fig 4 seems to be missing.

It is not clear to me what is original data presented for the first time and what is data taken from other sources. Attributing the source of the data would make the paper easier to follow.

In Fig 6, is the location of the mRNA binding groove known and can it be indicated on the structure?

Minor comments

Is it known why the mutation rate is higher in Dd2 than 3D7?

In Fig 5A it is shown that parasite resistance is more likely to occur in naïve donor cells rather than endemic donor cells. A possible explanation given is that the endemic cells may have resistance polymorphisms and/or innate immune responses may be present. Was parasite growth without inhibitors in naïve versus endemic blood ever performed to determine if parasites grow slower in endemic blood?

Table S7. Is there value in showing this without showing a model of how the analogues for more poorly into Eef2?

Fig S2E. What are the little block dots showing?

Line 289 Typo 'Y186N7'

Reviewer #4 (Remarks to the Author):

In their manuscript “Propensity of Selecting Mutant Parasites for the Antimalarial Drug Cabamiquine”, Stadler et al present a range of interesting data from a number of studies involving multiple different assays attempting to quantify malaria parasites’ acquisition of resistance mutation in the face of selection pressure from the drug Cabamiquine. They then attempt to derive an integrated modelling framework with which to bring these sources of data together to harness the combined power of the data to provide quantitative and comparable measures of the propensity for resistance allele to be generated, with which to gain insight into the likely trajectory of resistance evolution of the parasite to this and other drugs subject to similar pre-clinical testing.

My areas of expertise focus upon transmission modelling and modelling the impact of drugs within a clinical rather than clinical setting, however I can see that such a framework would be really valuable and more broadly - as a non-specialist- I found the summary of how different pieces of the puzzle fit together in terms of different assays and approaches sufficiently interesting to warrant publication in a wider-interest journal such as Nature Communications. There are some also some very nice illustrative graphics that really helped to improve my understanding of the field, though in places attention could be made to making terminology and overall aims of the approach more accessible to those less couched in the specifics of the biology.

However, taken at face value I’m afraid the framework doesn’t achieve what it sets out to do – the approaches to calculating key quantities such as resistance frequency produce widely different results depending on the assay, which the mathematical model cannot explain (Figure 5). The authors suggest this could be due to inherent differences in the assays and experimental conditions (“This may be due to differences in assay design between the MIR and recrudescence approaches, experimental conditions, or differences resulting from the use of RBCs from malaria-experienced or naïve donors across the two assay platforms”) but this does lead to a question about the wider value of the work (i.e. what can this approach actually be used to inform). Moreover, the modelling framework is, to my mind, far too simplistic and in places appears to be erroneously applied which makes me question the extent to which the differences are introduced by the methodology of the manuscript rather than something more inherent. As a result I would not recommend this article for publication without a major methodological overhaul. Even then, the underlying benefit of the approach if it continues to produce divergent estimates according to assay would need to be articulated better.

The model under-pinning the modelling component of the analysis is very simplistic – a simple model of homogenous exponential growth over successive generations with no link to the underlying biology of the parasite (e.g. variation in the offspring distribution, particularly when at low numbers of parasitaemia or the fixed 48 hour cycle of the parasite which likely introduce high degrees of stochasticity you would see if measuring multiplication as a function of time or treating at a threshold level of parasitaemia as many of the more complex in vivo studies do). There is no competition for resources between WT and resistant parasites, though this is described as an important factor in the discussion. Single values are considered for all parameters selected from relatively sparse data rather than considering a range equivalent to their inherent uncertainty.

The authors then rely on the assumption that the frequency of resistance has reached equilibrium – in the early stages of a stochastic process of replication this is clearly a strong assumption and is clearly contradicted by the data where some assays with the same initial conditions yield recrudescence whilst others do not. If they were at equilibrium all would recrudescence or none would or there would have to be other processes at play not captured in the model or inferential framework. A few simple simulations of the underlying model from same initial conditions would show the ranges of frequencies possible and I would encourage the authors to actually code up the model and provide the code for this – comparing what happens in multiple experiments in silico vs the equilibrium solution. They would see a range of frequencies ranging from 0 to higher frequencies associated with mutation early in the process.

Line 139 of SI: The authors assume the probability of no resistant parasites is a binomial distribution with n = number of initial inoculum. This would rely upon an assumption of no further replication from this initial n . This highly unlikely to be true so yields an under-estimate of the number of effective WT parasites – without understanding these experiments too well I would assume the replication of WT is quite limited if killing effect is much higher than PMR (though would need to be reassured this is the case) but this is an original sin which then gets compounded later on when different levels of drug parasitic effects are considered.

Line 162-165 of SI: “We then use the simplifying assumption that there is exactly one resistant mutant present at the time of treatment in mice and patients with recrudescence of parasites”. I can’t for the life of me get my head around this assumption at all as it seems completely counter to what the model would produce and seems to be a way to just be able to generate a result without considering the worthiness of the assumption and knock-on impact. In reality I would guess this would reflect an orders of magnitude under-estimate of frequency depending on assay which could potentially explain why the estimates from this approach are so much lower than the model generates. Again I encourage the authors to simulate output from their model, make the simplifying assumption of there only being 1 resistant parasite (regardless of how many there actually are in the simulation) and see what estimates of frequency this returns.

Line 218-221 – “If all parasites at treatment are wild type and a fraction λ of the parasites is not affected by the drug and replicates” – the key here is in the word ‘replicates’ here the framework is again assuming there are no more WT parasites post-drug introduction when in fact, depending on the drug properties – there could easily be further exponential growth due to further replication if the drug doesn’t get the PMR below one. This leads to the slightly bizarre suggestion in Figure 5B that an assay with a drug with no efficacy has only a 60% chance of having a resistance prior to treatment (i.e. a 40% change of generating resistance) – in reality, without any mechanism for competition, resistance mutations would always appear post-treatment with a drug with no efficacy in this model, if not already present already, just due random chance as parasite numbers head toward infinity. Indeed they would appear with mathematical certainty for any drug effect below the MIC to reduce $PMR < 1$. Where MIC is achieved this would need to be altered according to the accumulative parasite multiplication associated with negative exponential growth until clearance.

I’m afraid I don’t believe either the model or the framework are fit for purpose and would encourage the authors to revisit their assumptions – beginning with applying their framework to simulated data from the model they are aiming to interrogate (and providing code so others can replicate and perhaps develop more nuanced within-host models), whilst taking care to pay attention to the validity of assumptions that they make.

Minor considerations:

Line 64 – typo think should read ‘clinical trials’ rather than ‘clinical tools’

Given readership of journal more detail for an audience specialising in resistance work would be very welcome – e.g. a longer description of how MIR studies are carried out (and why etc.) Also some introduction to the mechanisms that likely drive resistance in these studies. Figure 1 is very nice though!

Similarly some concepts and abbreviations aren’t defined prior to their use (EC_{50}/nM – names of different strains and their significance..) which makes life harder for non-specialist – simple explanations of how to interpret these numbers would be useful too.

Line 130-3 – ‘for 50 out of 52 131 clinical isolates and 49 out of 52 control 3D7 cultures (3D7_MM and 3D7_FS), indicating no significant 132 difference in the propensity of field isolates to produce resistant parasites’ – this statement isn’t false but with this underlying proportion (approx. 4%) and sample size you would have to have a very big difference to produce significance so not sure it rules out that there aren’t important differences...

Early part of the results section – description of available data in paragraph from line 91 jumps between a range of different assays and back again very rapidly. Suggest start by laying out the data and experiments from each assay type (e.g. I can’t find how many/what % of the NSG mouse experiments yielded resistance phenotypes) – then by a section explaining how they potentially all fit together towards an integrated modelling framework.

Lines 321-322 – “Independent of the strain (field isolates or the reference 322 3D7 strain), the selection of mutants was much more challenging in field-based in vitro recrudescence 323 assays” – relative to what? Hard to work out which part of the analysis this is calling back to..

Reviewer #5 (Remarks to the Author):

This manuscript represents new data in sequencing emergent mutations in the eIF2 gene after exposure to carbimiquine, amalgamated with previously reported mutations (supp table S1). The data, when put together spans in vitro, mouse treatment selected, and human clinical trial selected data. This is the first time such rates have been compared, lending great value to this manuscript.

The data and methods are well presented and relatively easy to understand.

My only major comment for improvement is can the authors state where such data would cause a halt in clinical development of a compound? Or at least discuss the parameters from such a study that would do so. And better discuss how much they would need a parasite reduction in the partner drug, do be able to move forward. And how much that would be compromised by some existing resistance in a partner drug, e.g. artemisinin resistance that results in slower clearing of parasites vs. mefloquine or other partners where classic changes in IC50s exist in SE Asia. I realize from what the authors say, that its a complex decision, that depends somewhat on fitness, that may affect transmissability, etc. But the decreased fitness they estimate from models does not prevent resistant mutants from being present in relatively small populations in clinically-relevant models, and selected out be drug treatment.

A minor point:

Current address: Weill Cornell Medical College, New York, NY 10021, USA is on the cover page, but the # symbol is not used for the authors.

REVIEWER COMMENTS

Reviewer #1 (Remarks to the Author):

The noteworthy results here concern a potential new antimalarial drug. Resistance to it easily evolves. There are three parts to the paper. The first uses several different approaches to estimate how readily and the mutations involved in resistance. That is definitely of interest if you care about the drug, and it is of interest that different experimental systems give similar ballpark estimates of the rate, and often (but not always) different mutations. The second part (lines 207-237) attempts to estimate whether the mutations arise before or after treatment. The third part is a compelling speculation on a key binding site (lines 239-256)

Response: We thank the reviewer for sharing insightful and constructive feedback. In the following points, we address the suggestions that helped us improve our manuscript.

So far as I can tell, there are no new data in this paper; what is new is the comparison across different experimental systems (Fig 1) and some new modeling. There is an argument that the paper is of considerable interest to malariologists and so perhaps could be in a more specialized journal. The interesting generality (beyond malaria) is really the compare and contrast of the different approaches to experimental evolution being brought to bear on the same drug (showing it is possible to attack the problem in different ways), but I am guessing the conclusions that flow from the compare and contrast are rather specific to this drug and bug (here, there is some convergence at the genetic level, but also some idiosyncrasies associated with particular experimental approaches). Nothing wrong with that specificity – for this drug, it's an important assessment for malariologists.

The paper is well written and easy to digest. There is enough detail to work out what was done etc.

The second part of the paper I find slightly odd and I would be inclined to drop that (minimally justify much better why is it important/interesting). I am not sure what it means to ask whether the resistant mutants were pre-existent or arose after drug treatment. If the null model is that the drug is not mutagenic, then I am not sure it really matters. For a fixed mutation rate and replication rate, it will boil down to how many mutations arise pre-treatment and how many arise post-treatment and that will just boil down to the population dynamics (when in the infection treatment starts, and how effective the drug is at reducing replication). It would be an interesting question to ask if the drug was mutagenic but that is not what is being done here. In the field of course the issue matters (can the drug kill all the bugs before a resistant mutant arises?), but that is not what is being calculated here.

Response: Regarding data novelty, Dd2 data (SI); in vitro field isolate work, some of the humanized mouse data, mathematical modelling, and structural analysis have not previously been published.

We agree with the reviewer at one level that the pre-existence vs emergence question has an intuitive answer. That is, unless a drug is mutagenic, the dynamics of growth before and after treatment will completely determine the likelihood of a resistant mutant emerging. However, as is indicated by reviewer 2 we feel this is not broadly established in the malaria field as it may be in, for example, the bacterial literature. Thus, we believe that demonstrating this point with an illustrative mathematical model is useful to make it more accessible to a broader audience. We have now attempted to address this motivation more clearly in the main text.

Lines 236-240: Importantly, the above data analysis has all assumed that a small subpopulation of resistant parasites was present at the time of treatment (pre-existence), and that no resistance emerged de novo after treatment. Here we extended our modelling to test the validity of this

assumption. We used a deterministic formulation (Fig. S3) to consider whether, with no pre-existing resistant mutants, it was likely that resistant mutants emerged de novo only after treatment had started to inhibit parasites.

Moreover, trying to answer that odd question requires the authors do some calculation of costs of resistance. Costs of resistance are not fixed and are highly dependent on the ecological conditions and the fitness measure involved (e.g. Huijben et al. Evolution, Medicine and Public Health 2018: 127-137). Even if the dynamic range of the fitness costs could be estimated properly in each experimental system, the extrapolation to field setting would be impossible. I don't think that really matters if you buy my argument that the question itself is not very interesting. The important point is that the spontaneous mutation rate is high however you measure it, and that consequently this drug will fail unless it is combined with at least one partner drug with a similar half-life and very different mode of action.

Response: We thank the reviewer for highlighting this limitation. We agree that fitness costs are not fixed and likely depend on the infection model. We note that this is a major simplification of the modelling, and that this model is not able to account for the many complexities of the different host environments. However, despite these simplifications, we find that the new stochastic model of multiplication and mutation of parasites predicts similar frequencies of resistance in each of the experimental settings, with the exception of the in vitro regrowth data (which we discuss in the results). Further, a sensitivity analysis has revealed that although fitness cost is important and is in the range of observed fitness costs in vitro, the model gives similar results. Thus, we feel that despite its limitations, the model is able to provide a general estimate of the endogenous frequency of resistant mutants in these systems and shows some comparability with the observed frequencies. Consequently, we have introduced a discussion of these limitations into the main text, as well as the suggested reference (Line 354-356):

Lines 354-368: A limitation of our stochastic model is that it assumes fitness costs are a fixed quantity over time and across infection models. However, the costs of resistance are likely not fixed and may be highly dependent on the ecological conditions¹. Although the model was relatively insensitive to the fitness cost parameter in the range tested (Fig. S1C), fitness cost will invariably be an important factor in more extreme scenarios where, for example, fitness is dramatically reduced. Our current model could not consider these factors, nor could we predict with certainty that these results would translate to a field setting - since fitness costs were not assessed in the different host environments or over time. Despite these simplifications we find the model accurately predicts the observed frequency of resistant mutants across different experimental systems, with the exception of the field isolate experiments. For the latter, host RBC-specific determinants and/or innate immune mechanisms may play a role in reducing the frequency of resistant mutants. Studies have demonstrated the critical interplay between the host and antimalarial drug treatment in terms of the clearance of drug-resistant parasites in patients^{2, 3}. We speculate that with the use of host cells from malaria-endemic residents, the fitness cost of resistant mutants may be lower. Further studies are merited to determine whether MIR values differ between field and laboratory settings.

The structural analysis (part 3, fig 6) is really interesting.

Response: We thank the Reviewer for his positive comment on the structural analysis (part 3, Fig. 6). To be in line with Reviewer #3, we have now presented this work in the introduction (line 73-75).

Reviewer #2 (Remarks to the Author):

This paper explores the emergence and growth of *Plasmodium falciparum* parasites exposed to the anti-malarial drug (cabamiquine) across a number of environments (in vitro, in humanized mice, in volunteer trials). The authors compare the genetics of resistant mutants identified across settings — finding some mutants are shared — and characterize their phenotypes. Using a common lab strain (3D7) and clinical isolates, in vitro, they find that most cultures are cleared under drug pressure and there is no significant difference in the propensity to evolve resistance between the lab strain and clinical isolates. Next, the authors estimate the frequency of resistant mutants at the start of treatment in the different contexts, finding pretty substantial variation (several orders of magnitude separating the lowest and highest frequencies). The authors then use a simple mathematical model to determine whether it is more likely that resistant mutants were already present at the start of treatment, or emerged afterwards, finding that the former was more likely, generally. Lastly, the authors build structural models of the parasite protein that is the drug's target, to investigate the potential impact of the identified mutations.

There is a lot happening in this manuscript and for me that obscures a little bit what the key messages are. For example, the structural part doesn't seem particularly well integrated (e.g., not really motivated in the introduction at all). The comparison of specific mutations across contexts is pretty interesting, but I'm not sure strong/novel the conclusion is that "in vitro work...may prove useful for the assessment of antimalarials early in their development". I am pretty excited about the work in between these two parts: understanding whether drug resistance comes from standing genetic variation in the parasite population at the time of treatment or de novo mutation once treatment is administered is important for tailoring treatment regimes and predicting the long-term outlook for any given drug. My main comments focus on this section, where I have some questions about the modeling and the language.

Response: We thank the reviewer for their supporting comments regarding the modelling of "pre-existence" vs "de novo" mutations after treatment.

1. A large concern I have about the model for the frequency of resistant mutants is that it is a deterministic model that focuses on equilibrium fractions. Of course, when mutants first emerge in a population their growth is going to be strongly subject to demographic stochasticity. I could imagine this is one reason why the model produces a much higher frequency of resistance than most of the other scenarios. Have you considered different formulations?

Response: We thank the reviewer for this thoughtful comment. As per the suggestions of this reviewer and those of reviewer 4, we have now developed a stochastic model of the emergence of resistance mutations, which accounts for individual agents replicating with a random number of progeny, and the possibility of stochastic elimination of individual resistant mutants that emerge. As suggested by the reviewer, this model predicts frequencies of resistant mutants in each system (new figure below) which are much more consistent with those observed in the experimental data. The model now not only agrees well with the data but highlights that much of the differences in resistant frequencies between in vitro, mouse and human results purely from expected differences in the design of these experimental systems, rather than any fundamental differences in the biology in these systems. Interestingly, this is all with the exception of the field isolate experiments, where different host cells (host cells from an endemic setting) were used and a much lower frequency of resistant parasites was observed, indicating that perhaps there is a fundamental difference in this system that is not explained by the different experimental setup. With this new

modelling, and the new conclusions of the work that result, we feel the manuscript's reach and interest have been substantially improved and we thank the reviewers for their suggestions.

2. In estimating the frequency of resistance from the data, I imagine the assumption that there were no resistant parasites in the inoculum if a well was negative (l. 408-410) could also lead to underestimating frequencies (again, due to demographic stochasticity). I assume this is standard protocol, though I wonder if the work of Helen Alexander (e.g., Alexander & MacLean PNAS 2020) might be relevant here? More broadly, I had a hard time understanding the details of these estimations.

Response: We thank the reviewer for another thoughtful comment. We agree it is possible that we will not detect all wells that contained resistant parasites as it is possible, for example, that one resistant parasite would be stochastically eliminated during outgrowth (especially due to the fitness cost likely present for such resistant mutants). Other factors also come into play, such as detection limits of the assay and sequencing for resistant mutants. The work of Alexander and MacLean is interesting and if we apply their estimate that there is a 5% chance that a well containing a resistant mutant doesn't grow to detection due to stochastic variability, we can provide a sensitivity analysis of the possible impact of this assumption. In doing this analysis, we find that the estimated resistance frequencies can be impacted by the possibility of false-negative wells, but the overall estimates are very similar and rather insensitive to this assumption (see below table). Nonetheless, we feel this is an important sensitivity analysis and now include this additional analysis in the supplementary material (Table S7).

Data	Fraction of mutants (95% CI) without false negative wells	Fraction of mutants (95% CI) with 5% false negative wells	Ratio
MIR in vitro (Dd2/Naïve donor cells)	1.33×10^{-7} ($7.78 \times 10^{-8} - 2.26 \times 10^{-7}$)	1.93×10^{-7} ($8.39 \times 10^{-8} - 4.46 \times 10^{-7}$)	1.46
MIR in vitro (3D7/Naïve donor cells)	2.11×10^{-8} ($7.27 \times 10^{-9} - 6.11 \times 10^{-8}$)	9.99×10^{-8} ($2.18 \times 10^{-8} - 4.59 \times 10^{-7}$)	4.74
Regrowth in vitro * (3D7/Endemic donor cells)	4.14×10^{-10} ($1.33 \times 10^{-10} - 1.28 \times 10^{-9}$)	4.36×10^{-10} ($1.40 \times 10^{-10} - 1.35 \times 10^{-9}$)	1.05
Regrowth in vitro (Field/Endemic donor cells)	2.63×10^{-10} ($6.59 \times 10^{-11} - 1.05 \times 10^{-9}$)	2.78×10^{-10} ($6.93 \times 10^{-11} - 1.11 \times 10^{-9}$)	1.05
Regrowth in vivo NSG (3D7/Naïve donor cells)	8.35×10^{-9} ($4.00 \times 10^{-9} - 1.74 \times 10^{-8}$)	1.33×10^{-8} ($3.98 \times 10^{-9} - 4.47 \times 10^{-8}$)	1.60
Regrowth in vivo VIS (3D7/Naïve individuals)	2.73×10^{-9} ($1.01 \times 10^{-9} - 7.33 \times 10^{-9}$)	2.92×10^{-9} ($1.07 \times 10^{-9} - 8.01 \times 10^{-9}$)	1.07

- Is it the case that the parasites are essentially being pooled within a setting? We are estimating the frequency of resistance across, e.g., all parasites in all hosts? If I've understood that correctly, is pooling ok given that parasites within a given host would then represent not totally independent subpopulations.

Response: We thank the reviewer for this important but more technical comment on the methodology of estimating viable parasite frequencies across multiple hosts (such as NSG mice vs humans). As the reviewer notes, our method assumes an independence of the parasite populations within a host. On the other hand, in reality of course parasites from within the same host species are likely more related to each other than between host species. However, here we will show that our method and a more complex method that accounts for inter-individual variability will yield virtually the same estimates. We favour the simpler method since it does not require an additional parameter (the growth rate of parasites) to be estimated, but we now explain more clearly in the methods the simplifying assumption of this method, and justify this choice more explicitly.

*Technical details: As the reviewer correctly interprets, we are making an assumption that every parasite across every individual has the same average probability of being a resistant mutant, and thus assume independence between parasites regardless of the individual they are from. The reason we have used this approach is because it turns out that because of exponential growth, the final replication cycle is by far the biggest contributor to any likely resistance mutations. Thus, we obtain similar results regardless of whether we assume a more complicated model and approximation process or use the more standard analysis. The advantage of the simpler approach is that it means that we are using the exact same analysis approach for analysing the MIR *in vitro* and the *in vivo* data. Our estimates do not rely on also estimating the parasite multiplication rate in each individual. Below we provide the demonstration of this concept for the NSG and VIS data.*

If we assume the inoculum in each animal/individual is N1 and parasites multiply each generation to produce M progeny (i.e. the PMR is M), and if parasites go through G generations before

treatment with probability, f , that any particular progeny is a resistant mutant, then the probability that a particular animal has no resistant mutants at the time of treatment is

$$P(\text{No resistance}) = (1 - f)^{N1 \times M} \times (1 - f)^{N1 \times M^2} \times (1 - f)^{N1 \times M^3} \times \dots \times (1 - f)^{N1 \times M^G}.$$

This equation will be dominated by the term $N1 \times M^G$, depending on how large M is (and in the humans and mouse system the PMR is much greater than 1). Thus, this will give a similar result to $P(\text{No resistance}) = (1 - f)^{N1 \times M^G}$, which reduces to the simplified equation we have used in our analysis and depends only on the total parasitemia at the time of treatment rather than the particular starting inocula and growth rates in each individual. It is important to note that this is only relevant in the experimental systems where parasites propagated in different individuals prior to drug exposure. In the MIR system the same parasite stock was serially diluted to initiate the experimental wells that were immediately exposed to drug pressure. Thus, in these experiments we have assumed independence, since there is no higher degree of relatedness between parasites in a given well and between wells. We assume they are replicate samples of the same stock. Therefore, we are able to use the simplified model above in this instance and it is not an approximation in that case but a standard method for analysing limiting dilution experiments. We have now explained the assumption of independence and its limitations in more detail:

Lines 446-448: We note that this method assumes independence in the probabilities of resistance among parasites from within the same well/mouse/individual compared with between individuals (see supplementary methods for details).

- I'm not clear on the effect of the assumption in l. 418-420 (and elsewhere) that exactly one resistant parasite is present at the start of treatment. The text explains that this is necessary for computing confidence intervals, but it's not immediately obvious to me how, or how else this assumption plays into the calculations. To me, it's weird for a take home message to be that most likely resistant mutants exist at the start of treatment, but to get to that conclusion we have to assume that there can only be 1.

Response: We thank the reviewer for pointing out the weakness of this assumption. This was a choice made for purely numerical optimisation reasons at the time, and yielded approximately the same estimates as the approach where we assumed that resistant parasites may be greater than or equal to 1. However, we agree this is confusing. Thus, we now report only the results from our revised method that assumes 1 or more resistant parasites were present in samples containing resistance, rather than strictly 1 parasite, and have updated the methods and sup. methods appropriately. For the reviewers benefit the old and new estimates of resistance, showing the effect of this change in method (as well as the inclusion of additional data for the NSG mice), are listed in the table below:

Data	Number of parasites to obtain 1 resistant mutant (95% CI)	
	Original submission	Revised and updated estimates in the current submission
MIR in vitro (Dd2/Naïve donor cells)	7.54×10^6 ($4.42 \times 10^6 - 1.28 \times 10^7$)	7.54×10^6 ($4.42 \times 10^6 - 1.28 \times 10^7$)
MIR in vitro (3D7/Naïve donor cells)	4.75×10^7 ($1.64 \times 10^7 - 1.38 \times 10^8$)	4.75×10^7 ($1.64 \times 10^7 - 1.38 \times 10^8$)
Regrowth in vitro * (3D7/Endemic donor cells)	2.49×10^9 ($8.03 \times 10^8 - 7.72 \times 10^9$)	2.42×10^9 ($7.80 \times 10^8 - 7.50 \times 10^9$)
Regrowth in vitro (Field/Endemic donor cells)	3.87×10^9 ($9.67 \times 10^8 - 1.55 \times 10^{10}$)	3.80×10^9 ($9.50 \times 10^8 - 1.52 \times 10^{10}$)
Regrowth in vivo NSG (3D7/Naïve donor cells)	1.64×10^8 ($5.28 \times 10^7 - 5.08 \times 10^8$)	1.20×10^8 ($5.74 \times 10^7 - 2.50 \times 10^8$)
Regrowth in vivo VIS (3D7/Naïve individuals)	4.51×10^8 ($1.69 \times 10^8 - 1.20 \times 10^9$)	3.67×10^8 ($1.36 \times 10^8 - 9.86 \times 10^8$)

3. I also struggled with the evolutionary language throughout the paper. Again, I quite like the question of whether there is standing genetic variation for resistance (it is “pre-existing”) or whether it emerges de novo following treatment, but the text largely avoids putting it that way.

- l. 69-70 “mutants...emerged directly as a result of drug pressure or ... as a result of background mutation rates”. This sort of gives the impression that the drug may be mutagenic. Ditto for line 210 “resistance emerged as a result of drug pressure”

Response: We thank the reviewer for this comment and have now added a clear note of the different possibilities early on, and we have changed the language throughout to indicate that we consider the possibilities that the mutants were pre-existent or emerged de novo after treatment.

Lines 69-73: Mathematical modelling of these data using both deterministic and stochastic models allowed us to estimate the frequency of parasite resistance to this compound across different in vitro and in vivo infection models. This modelling also predicted the likelihood that these mutants either emerged de novo following drug treatment or alternatively were likely present at the time of treatment (pre-existent).

- l. 307 “random mutation, rather than selection pressure, could explain the presence of escape mutants”. I realize I am being a pedant here, but in either case mutation is necessary!

Response: We agree that this was not clearly worded. Our revised text now reads:

Lines 324-328: This was consistent with previous modelling in HIV, which reported that resistance mutants were more likely to emerge from random mutation prior to strong selection pressure (in the case of HIV from immunity) than from the same random mutation processes after selection

pressure has emerged, mainly because of the large viral load prior to immune-mediated selection pressure⁴.

Minor comments.

- l. 80-81 – I was a bit confused by this language because it could be interpreted as meaning the identical mutations (rather than identical residues, but different amino acids), of which none is shared across all three settings.
- Maybe not important for most reader of this paper, but I don't think EC₅₀ is defined anywhere.

Response: We have now introduced the definition in line 94 as half maximum parasite growth inhibition.

- l. 212, should this be a reference to Figure 5A?

Response: The numbering of figures has been adjusted in the manuscript (5 figures in total).

- l. 294-296, do you mean “load at treatment” rather than “inoculum at treatment”? (or perhaps simply “inoculum”?)

Response: We corrected to “parasite numbers” in the manuscript.

- l. 299-301, the phrase “in a manner dependent on drug pressure” is doing a lot of work here and “could predominate” rather than “would predominate” seems more appropriate. Having said that, could one (roughly?) quantify the conditions under which the lower-level resistant mutants would predominate? Or perhaps not if the fitness costs are determined in a no drug environment — one would need to know the relative fitness of the different resistant mutants at different drug doses?

- l. 443-444, make explicit here that another assumption is that all resistance mutations are identical phenotypically? (equal p_r?)

Response: We have now added our assumption that all resistance mutations have the same PMR and thus the same fitness cost to the supplementary methods where we describe the mathematical models and parameters in detail.

Lines 214-216 in the Supplement: In this model, we also assume that all resistant parasites have the same PMR pr and thus the same fitness cost for resistance.

Lines 277-281 in the Supplement: Fitness cost for resistant parasites: We assume that all resistant parasites have a 7% reduction of their PMR compared to sensitive parasites.

Reviewer #3 (Remarks to the Author):

Review of ‘Propensity of Selecting Mutant Parasites for the Antimalarial Drug Cabamiquine’ by Stadler et al.

Major Comments

Cabamiquine (Cab) is a promising new antimalarial drug undergoing preclinical evaluation which targets eukaryotic elongation factor 2 (EEF2) required for parasite protein synthesis. Under drug selection parasites resistant to Cab can arise and these can have mutations at 11 amino acid

positions in the protein and some of these mutations are shared between parasites grown in a dish, in a mouse or a human volunteer. Modelling predicts that these mutations were likely already present in the parasite populations prior to resistance selection rather than arising after drug after selection. Structural modelling suggests these mutations cluster around EEF2's mRNA binding groove. Overall, this was an interesting paper although I feel it lacked clarity in places and could have been better presented.

Response: We have improved the manuscript based on the suggestions thanks to the reviewer's insightful and constructive feedback.

Fig 4 seems to be missing.

Response: This typo has been corrected. The manuscript contains only 5 figures.

It is not clear to me what is original data presented for the first time and what is data taken from other sources. Attributing the source of the data would make the paper easier to follow.

Response: We are grateful for this comment. Regarding data novelty, Dd2 data (SI); in vitro field isolates work, some NSG mouse data, mathematical modelling, and structural analysis have previously not been published.

In Fig 6, is the location of the mRNA binding groove known and can it be indicated on the structure?

Response: The shallow grooves detected suggest as potential mRNA binding sites in analogy to work performed by Zerio et al.⁵ on eELF4 with a structurally similar inhibitor (quinoline-4-carboxylic acid with electron withdrawing group on 6-position and substitution at 2 position). In that study, the authors predicted with molecular docking that the inhibitor bound to the RNA-binding groove of eIF4A.

Minor comments

Is it known why the mutation rate is higher in Dd2 than 3D7?

Response: Our original data for Dd2 was generated from 10 studies versus 4 for 3D7, and we thus attribute greater confidence for the former, as shown by a smaller range of frequencies shown in Fig. 4A. An earlier report in PNAS⁶ had suggested that Dd2 WAS 10 to 100-fold mutable than 3D7. Four studies since then, however, have shown comparable rates. From those studies, we calculated median mutation rates of 1.05×10^{-9} for 3D7 and 1.78×10^{-9} for Dd2. These are listed in our revised Table S10. Our new stochastic model show comparable MIR values for both strains, as is now indicated in our revised Fig. 4 and Table S6.

In Fig 5A (Fig4A) it is shown that parasite resistance is more likely to occur in naïve donor cells rather than endemic donor cells. A possible explanation given is that the endemic cells may have resistance polymorphisms and/or innate immune responses may be present. Was parasite growth without inhibitors in naïve versus endemic blood ever performed to determine if parasites grow slower in endemic blood?

Response: Unfortunately, due to the logistics of obtaining blood from naïve individuals in a malaria-endemic field setting, we could not test the difference between naïve and endemic blood. Nevertheless, the P. falciparum 3D7 parasite's growth differs from donor to donor when we use blood from different donors in endemic countries (Figure below). This difference leads us to believe that various donors can affect parasite growth.

Table S7. Is there value in showing this without showing a model of how the analogues fit more poorly into Eef2?

Response: We thank the reviewer for pointing this out. We have now added docking poses to the table along with comments that could explain the observed EC₅₀.

Fig S2E. What are the little black dots showing?

Response: The black dots are highlighted by dummy atoms detected by SiteMap for the binding site.

Line 289 Typo 'Y186N7'

Response: This has now been corrected in the revised manuscript.

Reviewer #4 (Remarks to the Author):

In their manuscript "Propensity of Selecting Mutant Parasites for the Antimalarial Drug Cabamiquine", Stadler et al present a range of interesting data from a number of studies involving multiple different assays attempting to quantify malaria parasites' acquisition of resistance mutation in the face of selection pressure from the drug Cabamiquine. They then attempt to derive an integrated modelling framework with which to bring these sources of data together to harness the combined power of the data to provide quantitative and comparable measures of the propensity for resistance allele to be generated, with which to gain insight into the likely trajectory of resistance evolution of the parasite to this and other drugs subject to similar pre-clinical testing.

My areas of expertise focus upon transmission modelling and modelling the impact of drugs within a clinical rather than clinical setting, however I can see that such a framework would be really valuable and more broadly - as a non-specialist- I found the summary of how different pieces of the puzzle fit together in terms of different assays and approaches sufficiently interesting to warrant publication in a wider-interest journal such as Nature Communications. There are some also some very nice illustrative graphics that really helped to improve my understanding of the field, though in places attention could be made to making terminology and overall aims of the approach more accessible to those less couched in the specifics of the biology.

However, taken at face value I'm afraid the framework doesn't achieve what it sets out to do – the approaches to calculating key quantities such as resistance frequency produce widely different results depending on the assay, which the mathematical model cannot explain (Figure 5). The authors suggest this could be due to inherent differences in the assays and experimental conditions (“This may be due to differences in assay design between the MIR and recrudescence approaches, experimental conditions, or differences resulting from the use of RBCs from malaria-experienced or naïve donors across the two assay platforms”) but this does lead to a question about the wider value of the work (i.e. what can this approach actually be used to inform). Moreover, the modelling framework is, to my mind, far too simplistic and in places appears to be erroneously applied which makes me question the extent to which the differences are introduced by the methodology of the manuscript rather than something more inherent. As a result I would not recommend this article for publication without a major methodological overhaul. Even then, the underlying benefit of the approach if it continues to produce divergent estimates according to assay would need to be articulated better.

The model under-pinning the modelling component of the analysis is very simplistic – a simple model of homogenous exponential growth over successive generations with no link to the underlying biology of the parasite (e.g. variation in the offspring distribution, particularly when at low numbers of parasitaemia or the fixed 48 hour cycle of the parasite which likely introduce high degrees of stochasticity you would see if measuring multiplication as a function of time or treating at a threshold level of parasitaemia as many of the more complex in vivo studies do). There is no competition for resources between WT and resistant parasites, though this is described as an important factor in the discussion. Single values are considered for all parameters selected from relatively sparse data rather than considering a range equivalent to their inherent uncertainty.

Response: We thank the reviewer for their comments and suggestions on the modelling, and as we will detail below, we feel that the reviewer's comments have led to a significant improvement in the manuscript. We first wish to clarify what is perhaps a slight miscommunication of what is being done in this paper and distinguish between two features of the work:

- (a) data analysis to estimate frequency of resistance from the data; and*
- (b) modelling of proliferation and mutation to predict the frequency of resistance mutation.*

It is important to note that our modelling in (b), both in the previous and current versions of the manuscript, were not used to estimate the frequency of resistance mutants (i.e. (a)). The frequency of resistance estimates from the data therefore are not dependent on the assumptions of the simplified model. The simplified model was aimed to be purely illustrative at showing what frequency of resistance mutants might be expected based purely on the mutation rate of the parasite. Having said this, we agree with the reviewer that to improve this work, and test more rigorously whether the estimates in (a) can be predicted by a model of proliferation and mutation, a stochastic model was required to take into account random variation, especially given the different experimental setups across the infection models.

Thus, following the very helpful suggestion of the reviewer, we have now developed and implemented a stochastic model of parasite proliferation and mutation. As requested, this model

includes: variation in the offspring distribution (Poisson), the fixed life-cycle length of the parasites, and a replicative fitness cost of resistant parasites. We have also now provided a sensitivity analysis of the impact of varying all parameters in the model around regions of uncertainty (Fig. S1).

This suggestion from the reviewer has dramatically improved the manuscript. The stochastic model has not only predicted frequencies of resistant mutants that strongly agree with those estimated from the data – they have highlighted predictable stochastic variation as a major reason for the different frequencies of resistance across in vitro (MIR), NSG and VIS infection models (Fig. 4A, also shown above). This is now a major improvement to the conclusions of the manuscript, and highlight that the only system where stochastic variation has been insufficient to explain the low frequency of resistant mutants is in the field isolate experiments.

Thus, this new stochastic model is now used throughout the main text, and the new results are now highlighted in the results.

The authors then rely on the assumption that the frequency of resistance has reached equilibrium – in the early stages of a stochastic process of replication this is clearly a strong assumption and is clearly contradicted by the data where some assays with the same initial conditions yield recrudescence whilst others do not. If they were at equilibrium all would recrudescence or none would or there would have to be other processes at play not captured in the model or inferential framework. A few simple simulations of the underlying model from same initial conditions would show the ranges of frequencies possible and I would encourage the authors to actually code up the model and provide the code for this – comparing what happens in multiple experiments in silico vs the equilibrium solution. They would see a range of frequencies ranging from 0 to higher frequencies associated with mutation early in the process.

Response: We thank the reviewer for this comment. As noted above, we agree that assuming parasites were at equilibrium frequencies when modelling resistant mutants has proved a poor assumption for predicting the NSG and VIS studies, whereas considering the stochastic nature of the process has allowed much more accurate predictions of the observed frequencies in these systems. However, we do wish to note a slight misconception that the frequency of resistance being at equilibrium does not necessarily mean that all animals or individuals will have resistant parasites. This is because when we indicate that the frequency of resistance is at equilibrium we mean that the expected (i.e. average) number of resistance parasites per parasite has approached steady state, and if you randomly sample from this population there is still the possibility that you will only select wild-type rather than resistant parasites (depending on the frequency of resistance and the number of parasites sampled). This is the premise of a limiting dilution assay (and the MIR assays used here), and the means by which the data here can be used to estimate the frequency of resistant mutants.

In addition, we show a comparison of the deterministic model of parasite multiplication and resistance mutation, with the new stochastic model (see Fig. S3). All our code, which will allow the reproduction of the frequency estimates from the data, the deterministic and stochastic models, and the stochastic simulations of the different experiments, are included in the submission and will be made publicly available on GitHub upon publication.

Line 139 of SI: The authors assume the probability of no resistant parasites is a binomial distribution with n = number of initial inoculum. This would rely upon an assumption of no further replication from this initial n . This highly unlikely to be true so yields an under-estimate of the number of

effective WT parasites – without understanding these experiments too well I would assume the replication of WT is quite limited if killing effect is much higher than PMR (though would need to be reassured this is the case) but this is an original sin which then gets compounded later on when different levels of drug parasitic effects are considered.

Response: The reviewer is correct. i.e. not in the modelling, but in the estimate of resistance frequency from the data, we assumed that all resistant mutants were present at the time of treatment when drug pressure was applied – as is always the case with MIR/limiting dilution type experiments. This is because drug is added at many times higher than the in vitro IC₅₀, and thus it is expected that nearly complete suppression of non-resistant parasites is achieved (see⁷ where this compound is shown to be highly potent at inhibiting parasites – hence why it was selected for use in clinical trials). The reviewer is nonetheless correct that we should test the assumption, and that is precisely one of the major motivations for us to include the modelling section of this work – i.e. to test whether it is reasonable for us to assume that resistance was likely present at the time of treatment or could have emerged following treatment. Though we note that for some this might seem obvious from first principles (e.g. Reviewer 1) – we felt it worth demonstrating with a modelling approach.

Our modelling shows that the stochastic model is very good at predicting that at the time of drug being added, the predicted frequency of resistance mutants is highly consistent with that estimated from the data. Thus, without invoking any other mechanism other than multiplication, mutation and fitness costs, we can predict that resistance was pre-existent prior to treatment and at frequencies consistent with those observed upon completion of each selection study. Further, running the model forward and applying drug, and assuming that the drug does not increase mutation rates (not mutagenic), we show that even if the drug effect is as low as inhibiting the growth of 80% of parasites (and we should note, the compound has been demonstrated to be much more inhibitory at these concentrations), then this would mean that parasites are more likely to have already been present at the time of treatment than to have emerged de novo after treatment (Fig. 4B). These two results strongly support the validity of the assumption that resistant mutants were pre-existent, as one might expect for a drug known to inhibit multiplication. We have now discussed the assumption that we are making when analysing the data and provide the arguments for this, and the possibility of a mutagenic property of the drug, which we have not excluded, but was not needed to explain the observations:

Lines 236-244: Importantly, the above data analysis has all assumed that a small subpopulation of resistant parasites was present at the time of treatment (pre-existence), and that no resistance emerged de novo after treatment. Here we extended our modelling to test the validity of this assumption. We used a deterministic formulation (Fig. S3) to consider whether, with no pre-existing resistant mutants, it was likely that resistant mutants emerged de novo only after treatment had started to inhibit parasites. For this, we assumed that when drug was administered, viable parasite numbers reduced by a certain fraction each replication cycle (the parasite reduction ratio, PRR) and on each generation of surviving parasites there was the probability of generating resistant mutants. To be conservative, we did not limit the number of cycles of parasite replication post-treatment⁸.

*Lines 328-3333: We have not explicitly considered the possibility that cabamiquine increased the mutation rate of the parasite in our analysis (i.e. that cabamiquine is mutagenic), based on the close agreement between the frequencies of resistance predicted by our stochastic model and those observed experimentally. The previously reported background rate of *P. falciparum* mutations and the number of replication cycles prior to treatment in each of these experimental systems was sufficient to explain the rates of resistance observed herein.*

Line 162-165 of SI: “We then use the simplifying assumption that there is exactly one resistant mutant present at the time of treatment in mice and patients with recrudescence of parasites”. I can’t for the life of me get my head around this assumption at all as it seems completely counter to what the model would produce and seems to be a way to just be able to generate a result without considering the worthiness of the assumption and knock-on impact. In reality I would guess this would reflect an order of magnitude under-estimate of frequency depending on assay which could potentially explain why the estimates from this approach are so much lower than the model generates. Again I encourage the authors to simulate output from their model, make the simplifying assumption of there only being 1 resistant parasite (regardless of how many there actually are in the simulation) and see what estimates of frequency this returns.

Response: We agree this was a problematic assumption, and apologise for including this choice without adequate justification. This assumption was used purely for numerical reasons after we determined that it would yield nearly identical results. However, the more generalised assumption should have been used in the manuscript. We have now amended this assumption and updated the analysis to consider the strictly correct possibility that there was ≥ 1 resistant parasite in all cultures/hosts that were confirmed with genetic testing to have resistant parasites (see also above response to reviewer 2).

Line 218-221 – “If all parasites at treatment are wild type and a fraction λ of the parasites is not affected by the drug and replicates” – the key here is in the word ‘replicates’ here the framework is again assuming there are no more WT parasites post-drug introduction when in fact, depending on the drug properties – there could easily be further exponential growth due to further replication if the drug doesn’t get the PMR below one.

Response: We thank the reviewer for identifying an assumption that should be tested in more detail. Previously, we assumed that a proportion of the WT parasites would survive treatment and go through a further single replicative cycle with the possibility of mutating at the same rate as prior to treatment. However, the reviewer makes an important point, that by only considering one replicative cycle after treatment we underestimate the possible resistant mutants that may emerge from multiple rounds of replication. Therefore, we have now generalised this analysis to look at the probability that a resistant mutant did not emerge prior to treatment but emerged post-treatment under different parasite reduction ratios (PRR), which is the PMR multiplied by the per generation drug effect and must be less than 1 for a drug to clear infection. Using this revised model that includes (as a conservative assumption) unlimited replication cycles under drug pressure we have updated Figure 4B. The updated figure shows that resistant mutants are very unlikely to emerge only after drug treatment compared to the probability of pre-existence of resistant parasites at the time of treatment, except where the drug fails to inhibit parasites considerably ($PRR < 5$). However, as has been shown this drug arrests parasite development at very low concentrations⁷, and a single dose led to cure in multiple human subjects.

This leads to the slightly bizarre suggestion in Figure 5B that an assay with a drug with no efficacy has only a 60% chance of having a resistance prior to treatment (i.e. a 40% chance of generating resistance) – in reality, without any mechanism for competition, resistance mutations would always appear post-treatment with a drug with no efficacy in this model, if not already present already, just due to random chance as parasite numbers head toward infinity. Indeed they would appear with mathematical certainty for any drug effect below the MIC to reduce $PMR < 1$. Where MIC is achieved this would need to be altered according to the accumulative parasite multiplication associated with negative exponential growth until clearance.

Response: We thank the reviewer for raising this discussion. An important point that we would like to make is that selection pressure is neither sufficient nor necessary to acquire resistant mutants, instead mutation is the key enabler. This is because parasite mutation occurs de novo at a constant rate, and without mutation no resistant mutants could emerge. Since all that is required for resistance to this compound is specific single nucleotide point mutations, there is a very high chance (given enough parasites) that a resistant mutant emerges in the absence of any treatment. The question is whether this is more or less likely before or after drug suppression of replication. In this analysis we show, as has been demonstrated in other contexts⁴, that the emergence of mutations that confer resistance is more likely before drug is added when there is lots of parasite expansion, compared to after treatment when most parasite replication is suppressed (in the instance of an effective drug). We have clarified this in the legend of our revised Fig. 4B.

Lines 203-207: (B) The probability of the emergence of resistant mutants after treatment compared with the probability that resistant mutants are already present at treatment for different parasite reduction ratios. The model predicts that the probability of resistant mutants already being present at the time of treatment is larger than the probability of resistant mutants emerging only after treatment for a parasite reduction ratio of 5 or more. These quantities were computed using the deterministic model (see Supplementary Materials and Methods).

I'm afraid I don't believe either the model or the framework are fit for purpose and would encourage the authors to revisit their assumptions – beginning with applying their framework to simulated data from the model they are aiming to interrogate (and providing code so others can replicate and perhaps develop more nuanced within-host models), whilst taking care to pay attention to the validity of assumptions that they make.

Response: In response to the helpful suggestions from the reviewer, we have substantially revised our modelling, which we feel has dramatically improved this aspect of our study.

Specifically, we have now performed the reviewer's requested analysis, i.e. we developed a stochastic model and ran 100,000 simulations of each experimental condition (see Figure 4A). This created 100,000 in silico repeats of the same experiments performed in real-life, matching experimental design details (such as starting the experiment with the same inocula as were used, propagating parasites for the same time prior to treatment, and using growth rates consistent with the data). We then applied our data analysis method as we did to the real data, to estimate the frequency of resistant mutants in each of these simulated experiments. This analysis revealed very strong agreement between the simulated experiments and the observed outcomes, with the exception of the field isolate experiment. We speculate that the reason for the latter disagreement may point to an interesting biological difference in resistance frequencies when red blood cells from a donor from an endemic setting is used instead of blood from naïve individuals. We feel with this new modelling work, the agreement between data and the modelling, and the new conclusions of these innovations, the work is substantially improved.

Minor considerations:

Line 64 – typo think should read 'clinical trials' rather than 'clinical tools'

Response: The sentence reads now as "However, how well these tools allow for the prediction of treatment failure due to resistance in clinical settings has been less well understood".

Given readership of journal more detail for an audience specialising in resistance work would be very welcome – e.g. a longer description of how MIR studies are carried out (and why etc.) Also some introduction to the mechanisms that likely drive resistance in these studies. Figure 1 is very nice though!

Similarly some concepts and abbreviations aren't defined prior to their use (EC50/nM – names of different strains and their significance..) which makes life harder for non-specialist – simple explanations of how to interpret these numbers would be useful too.

Response: We have now included a definition of EC₅₀ (line 101). The P. falciparum 3D7 strain is the reference strain in the malaria field.

Line 130-3 – ‘for 50 out of 52 131 clinical isolates and 49 out of 52 control 3D7 cultures (3D7_MM and 3D7_FS), indicating no significant 132 difference in the propensity of field isolates to produce resistant parasites’ – this statement isn't false but with this underlying proportion (approx. 4%) and sample size you would have to have a very big difference to produce significance so not sure it rules out that there aren't important differences...

Response: We agree with the reviewer. The lack of significance may very likely be due to the low sample size compared to the low underlying proportion of isolates with resistant parasites. We have now commented on the limitation of low numbers in this work in the discussion:

Lines 339-343: Although we did not observe differences in the number of field isolates (2/52) that developed resistance compared with the frequency of the 3D7 laboratory strain (3/52), the low numbers of resistant lines limited the power to observe a difference between these groups. Thus, we cannot exclude the possibility that propensities for resistance may differ between these field and laboratory strains.

Early part of the results section – description of available data in paragraph from line 91 jumps between a range of different assays and back again very rapidly. Suggest start by laying out the data and experiments from each assay type (e.g. I can't find how many/what % of the NSG mouse experiments yielded resistance phenotypes) – then by a section explaining how they potentially all fit together towards an integrated modelling framework.

Response: This has now been addressed in Table S3.

Lines 321-322 – “Independent of the strain (field isolates or the reference 322 3D7 strain), the selection of mutants was much more challenging in field-based in vitro recrudescence 323 assays” – relative to what? Hard to work out which part of the analysis this is calling back to..

Response: We have now clarified in Line 390: “Independent of the strain (field isolates or the 3D7 reference), the selection of mutants was much more challenging in field-based in vitro recrudescence assays compared to MIR studies.”

Reviewer #5 (Remarks to the Author):

This manuscript represents new data in sequencing emergent mutations in the eIF2 gene after exposure to carbimiquine, amalgamated with previously reported mutations (supp table S1). The data, when put together spans in vitro, mouse treatment selected, and human clinical trial selected data. This is the first time such rates have been compared, lending great value to this manuscript.

The data and methods are well presented and relatively easy to understand.

Response: We thank the reviewer for their appreciation of our work. We implemented these comments in the revised manuscript as suggested.

My only major comment for improvement is can the authors state where such data would cause a halt in clinical development of a compound? Or at least discuss the parameters from such a study that would do so. And better discuss how much they would need a parasite reduction in the partner drug, do be able to move forward. And how much that would be compromised by some existing resistance in a partner drug, e.g. artemisinin resistance that results in slower clearing of parasites vs. mefloquine or other partners where classic changes in IC50s exist in SE Asia. I realize from what the authors say, that its a complex decision, that depends somewhat on fitness, that may affect transmissibility, etc. But the decreased fitness they estimate from models does not prevent resistant mutants from being present in relatively small populations in clinically-relevant models, and selected out be drug treatment.

Response: We have now discussed this point:

Line 319-321: Thus, for an individual with 10^{12} parasites, a partner drug would be required to bring parasite loads below 10^8 (preferably far below this level) to reduce the risk of drug resistance and avoid recrudescence.

A minor point:

Current address: Weill Cornell Medical College, New York, NY 10021, USA is on the cover page, but the # symbol is not used for the authors.

Response: This symbol has been associated to Josefina Striepen.

References

1. Huijben S, Chan BHK, Nelson WA, Read AF. The impact of within-host ecology on the fitness of a drug-resistant parasite. *Evol Med Public Health* **2018**, 127-137 (2018).
2. Austin DJ, White NJ, Anderson RM. The dynamics of drug action on the within-host population growth of infectious agents: melding pharmacokinetics with pathogen population dynamics. *Journal of theoretical biology* **194**, 313-339 (1998).
3. Wernsdorfer WH. Drug resistant malaria. *Endeavour* **8**, 166-171 (1984).
4. Lee HY, *et al.* Modeling sequence evolution in acute HIV-1 infection. *Journal of theoretical biology* **261**, 341-360 (2009).
5. Zerio CJ, *et al.* Discovery of an eIF4A Inhibitor with a Novel Mechanism of Action. *J Med Chem* **64**, 15727-15746 (2021).
6. Rathod PK, McErlean T, Lee P-C. Variations in frequencies of drug resistance in Plasmodium falciparum. *PNAS* **94**, 9389-9393 (1997).
7. Baragana B, *et al.* A novel multiple-stage antimalarial agent that inhibits protein synthesis. *Nature* **522**, 315-320 (2015).
8. Sanz LM, *et al.* P. falciparum in vitro killing rates allow to discriminate between different antimalarial mode-of-action. *PLoS One* **7**, e30949 (2012).

REVIEWERS' COMMENTS

Reviewer #2 (Remarks to the Author):

This is a revised version of a previously-reviewed manuscript. I appreciate the authors' responses to that earlier round of review. I think the addition of the stochastic model, in particular, has strengthened the manuscript.

I have a few thoughts that build off of the comments of the other reviewers.

I find Reviewer 1's skepticism of the model compelling, i.e., that whether a resistance mutation pre-existed in the infection or arose de novo once treatment started will come down to the population dynamics / time until treatment is sought, etc — i.e., things that (presumably) can't easily be controlled. The response of the authors here highlights the value of bringing the idea of these two sources of resistance to the forefront for the malaria field, but it would be nice to see some more clear/strong statements about what to do with the fact that resistance is likely to be pre-existing (kind of along the lines of what reviewer 5 was asking for). Am I correct in interpreting this as bad news for the drug?

Relatedly, lines 315-321 talk about tailoring a partner drug to account for the frequency of mutants resistant to cabamiquine, but I'm confused about this. If combinations were used sequentially, so that the partner drug was used first to reduce the parasite population below 10^8 or "preferably far below this level" before cabamiquine was applied, then this makes sense to me. But used simultaneously, drug treatment will enrich for any single or doubly-resistant mutants, so getting the population to 10^8 might not be good enough?

Minor points from the main text.

- what does the asterisk mean in Figure 4A ("Regrowth in vitro*")
- l. 366-368 "...with the use of host cells from malaria-endemic residents, the fitness cost of resistant mutants may be lower" — Is this a typo? Given that the estimated frequency of resistance is much lower in these cases, is the interpretation that the fitness cost might be higher?
- l. 485, will a link to the GitHub repository be provided?

Minor points from the Supplementary Materials & Methods

- add a reference to "f" somewhere in line 135? This line sounds like it's defining f, but f isn't defined until l. 151.

- l. 169-170, is this reference to the SM&M a typo?

- l. 191-192, I'm confused about what this line is describing. This is... not a confidence interval?

- About the simple model: it's a bit strange to me that a given parasite in generation $n-1$ can either mutate or replicate to produce p progeny parasites. In my head, this $s(n)$ equation should be $p(1-m)s(n-1)$. [For $r(n)$, maybe the second term is fine given the description of the mutation rate "1 mutant emerging from a parent population..."]. I don't actually think this would make a practical difference — the deviations between those model variants are small, I think — but I'm curious if this choice can be better explained.

Reviewer #5 (Remarks to the Author):

Thank you for addressing my points.

Reviewer #2 (Remarks to the Author):

This is a revised version of a previously-reviewed manuscript. I appreciate the authors' responses to that earlier round of review. I think the addition of the stochastic model, in particular, has strengthened the manuscript.

Response: We thank the reviewer for the positive feedback on the manuscript and the additional modelling work.

I have a few thoughts that build off of the comments of the other reviewers.

I find Reviewer 1's skepticism of the model compelling, i.e., that whether a resistance mutation pre-existed in the infection or arose de novo once treatment started will come down to the population dynamics / time until treatment is sought, etc — i.e., things that (presumably) can't easily be controlled.

Prior response: We agree with the reviewer at one level that the pre-existence vs emergence question has an intuitive answer. That is, unless a drug is mutagenic, the dynamics of growth before and after treatment will completely determine the likelihood of a resistant mutant emerging. However, as is indicated by reviewer 2 we feel this is not as broadly established in the malaria field as it may be in, for example, the bacterial literature. Thus, we believe that demonstrating this point with an illustrative mathematical model is useful to make it more accessible to a broader audience. We have now attempted to address this motivation more clearly in the main text.

The response of the authors here highlights the value of bringing the idea of these two sources of resistance to the forefront for the malaria field, but it would be nice to see some more clear/strong statements about what to do with the fact that resistance is likely to be pre-existing (kind of along the lines of what reviewer 5 was asking for). Am I correct in interpreting this as bad news for the drug?

Response: We agree with the reviewer that further interpretation of our results would be helpful for the field. Resistant mutants have emerged to nearly all antimalarial drugs, perhaps with the exception of lumefantrine, and thus it is critical to assess propensity of drugs to develop resistance. The novelty of our work is to provide guidance on how well in vitro and pre-clinical data on the propensity (i.e. frequency) of resistant mutants translates to clinical settings. We find, in general that resistance frequency is very well explained by the pathogen dynamics, mutation rates and the number of point mutations required to confer resistance. Thus, any antimalarial compound where resistance can be conferred by single point mutations is likely to have similar frequencies of resistance to those estimated herein for cabamiquine. This highlights the need for cabamiquine, like all antimalarials, to be used in combination with other agents (as is recommended by the WHO [[https://doi.org/10.1016/0169-4758\(96\)10055-7](https://doi.org/10.1016/0169-4758(96)10055-7)]), or to be used primarily in setting where there are low parasite densities (and so a low chance of a resistant mutant being present), e.g. as chemoprevention. We have now expanded our discussion to better explain the interpretation of our results for the field.

Our analysis highlighted that it is more likely that cabamiquine-resistant mutants were pre-existent rather than that they emerged de novo after treatment at frequencies approaching 1 resistant parasite per approximately 2.4×10^7 parasites given enough time for parasites to achieve high loads. We have not explicitly considered the possibility that cabamiquine increased the mutation rate of the parasite in our analysis (i.e. that cabamiquine is mutagenic), based on the close agreement between the frequencies of resistance predicted by our stochastic model and those observed experimentally. Instead, the previously reported background rate of *P. falciparum* mutations and the number of replication cycles prior to treatment in each of these experimental systems was sufficient to explain the rates of resistance observed herein. This was consistent with previous modelling in HIV, which reported that resistance mutants were more likely to emerge from random mutation prior to strong selection pressure (in the case of HIV from immunity) than from the same random mutation processes after selection pressure has emerged, mainly because of the large viral load prior to immune-mediated selection pressure¹. Together, this result suggests that any antimalarial compound where moderate or high-grade resistance can be conferred by any one of approximately eleven single point mutation is likely to have similar frequencies of resistance to cabamiquine. This highlights the need for using cabamiquine in combination therapies (as is recommended for all antimalarial drugs ([https://doi.org/10.1016/0169-4758\(96\)10055-7](https://doi.org/10.1016/0169-4758(96)10055-7))) or in settings where low parasite burdens are expected, for example as chemoprevention.

Relatedly, lines 315-321 talk about tailoring a partner drug to account for the frequency of mutants resistant to cabamiquine, but I'm confused about this. If combinations were used sequentially, so that the partner drug was used first to reduce the parasite population below 10^8 or "preferably far below this level" before cabamiquine was applied, then this makes sense to me. But used simultaneously, drug treatment will enrich for any single or doubly-resistant mutants, so getting the population to 10^8 might not be good enough?

Response: As the reviewer indicates, combinations are not administered sequentially. The reviewer then suggests that simultaneous use of two compounds will "enrich for any single or doubly-resistant mutants". Although it is true in principle that any pre-existent doubly resistant parasites would be enriched by combination therapy, the reasoning behind the WHO recommendations for use of combination therapies is that such doubly resistant parasites are far less likely and thus will occur at much lower frequencies. For example, if there is a 4.13×10^{-8} probability of a parasite being resistant to one drug, and a probability of 4.13×10^{-8} that a parasite will be resistant to a second drug (and assuming the drugs are different enough in their mode of action that these probabilities are independent) then there is a 1.7×10^{-15} probability of multi-drug resistance. In the case of cabamiquine, it was observed that partnering the drug with pyronaridine abrogated treatment failure (Rottmann et al AAC 2020). Singly resistant parasites will always be susceptible to one of the drugs and thus should not be enriched with combination treatment.

Nonetheless, we do agree with the reviewer that the choice of partner drug is more complicated than our simplistic discussion point indicates. This is because the frequency of pre-existent resistant parasites is not the only consideration. For example, one must consider drug interactions in order to both optimize effectiveness and reduce the risk of resistance², and choosing appropriately matched drug half-lives is likely important to prevent resistance emergence and spread^{3,4,5}. We have now discussed these complexities in more detail and explained that the 10^8 parasites are only indicative of the required potency of a potential partner drug but other considerations are also critical:

Lines 359-383: The previously reported background rate of *P. falciparum* mutations and the number of replication cycles prior to treatment in each of these experimental systems was sufficient to explain the rates of resistance observed herein. In selecting a partner drug during the development of antimalarial combinations, there are a number of important factors to consider, such as the half-lives of each drug^{3, 4, 5}, drug-interactions and how these will impact on overall effectiveness and propensity of the combination to resistance². In addition, one element to consider is the number of surviving parasites that would need to remain susceptible to the second drug, in the case that the partner drugs do not interact. Using the data from the 12 mg/kg dose in NSG mice, which corresponds ~to 500 mg (free base) of cabamiquine used for the human dose, we estimated a frequency of about 1 resistant mutant per 10⁸ parasites. Thus, for an individual with 10¹² parasites, the partner drug would contribute in reducing the number of parasites to <10⁸ (preferably far below this level) and eliminate cabamiquine-resistant parasites to avoid recrudescence.

Minor points from the main text.

- what does the asterisk mean in Figure 4A (“Regrowth in vitro*”)

Response: We have added an explanation in the figure legend to explain the asterisk:

*** for the 3D7 in vitro regrowth assay, we assumed that the parasite number at the time of treatment for each culture is the mean parasite number from the cultures for which the parasitemia at treatment was known, see Supplementary Materials and Methods.**

- l. 366-368 “...with the use of host cells from malaria-endemic residents, the fitness cost of resistant mutants may be lower” — Is this a typo? Given that the estimated frequency of resistance is much lower in these cases, is the interpretation that the fitness cost might be higher?

Amended. This should have read “higher”.

- l. 485, will a link to the GitHub repository be provided?

Response: The data and code is now available on GitHub (<https://github.com/estadler/cabamiquine>) and on Zenodo (<https://doi.org/10.5281/zenodo.8111676>).

This was also updated in the Materials and Methods section and the Data and materials availability statement.

Minor points from the Supplementary Materials & Methods

- add a reference to “f” somewhere in line 135? This line sounds like it’s defining f, but f isn’t defined until l. 151.

Response: Amended.

- l. 169-170, is this reference to the SM&M a typo?

Response: The references to the Supplementary Materials and Methods in lines 169-170 is not a typo. Though we have also included some information on the estimation of the frequency of

resistant mutants in the Materials and Methods section, the details of these estimates are still in the Supplementary Materials and Methods and are meant to be referenced here.

- l. 191-192, I'm confused about what this line is describing. This is... not a confidence interval?

Response: There was a part of the Supplementary Materials and Methods missing here. This was indeed not a confidence interval but concerned the estimation of the frequency of resistant parasites without the assumption that all cultures that were negative did not contain resistant parasites. This section was now added again.

Line 243: Estimation of the frequency of mutants allowing for false negatives

In the above estimation of the frequency of resistant mutants, we assumed that there are no false negatives, i.e. a negative well is assumed to have no resistant mutants. However, it is possible that not all resistant parasites are detected and some wells are falsely categorized as negative wells without resistant parasites. This could be due to, for example, stochastic elimination of a low number of resistant parasites during outgrowth or detection limits of the assay and sequencing for resistant mutants. Thus, to study the importance of assuming that there are no false negatives, we also estimated the frequency of resistant mutants assuming that 5% of wells with at least one resistant parasite are false negatives⁶ and 95% of wells with at least one resistant parasite are correctly identified as positive wells. Thus, the probability of a positive well is given by

$$P(\text{positive well}) = 0.95 \times P(\geq 1 \text{ resistant parasite}) = 0.95 \times (1 - B(0 | n, f)),$$

where B denotes the binomial distribution, n the number of parasites at treatment, and f the fraction of resistant parasites. Using this probability for a positive well, we estimate the fraction of resistant parasites in the different data sets as outlined above and compare the estimate with our previous estimate in which we assume that there are no false negative wells (see Table S7).

- About the simple model: it's a bit strange to me that a given parasite in generation $n-1$ can either mutate or replicate to produce p progeny parasites. In my head, this $s(n)$ equation should be $p(1-m)s(n-1)$. [For $r(n)$, maybe the second term is fine given the description of the mutation rate "1 mutant emerging from a parent population..."]. I don't actually think this would make a practical difference — the deviations between those model variants are small, I think — but I'm curious if this choice can be better explained.

Amended: We have now explained that this was chosen to better approximate how these parameters are measured in the literature.

Line 288: This model parameterization was chosen so that the parameters p and m approximate, as closely as possible, the parameters of PMR and mutation rate as they are measured in the literature (i.e. the PMR is estimated as the ratio of total parasites between generations and the mutation rate is estimated per parent population).

As the reviewer notes, alternative parameterizations are possible, but because of the small probability of mutation compared to PMR, these choices have no material impact on the results.

Reviewer #5 (Remarks to the Author):

Thank you for addressing my points.

Response: We thank the reviewer for the positive feedback on the revised manuscript.

References

1. Lee HY, *et al.* Modeling sequence evolution in acute HIV-1 infection. *Journal of theoretical biology* **261**, 341-360 (2009).
2. Chait R, Craney A, Kishony R. Antibiotic interactions that select against resistance. *Nature* **446**, 668-671 (2007).
3. Stepniewska K, White NJ. Pharmacokinetic determinants of the window of selection for antimalarial drug resistance. *Antimicrob Agents Chemother* **52**, 1589-1596 (2008).
4. Khoury DS, Cao P, Zaloumis SG, Davenport MP, The Interdisciplinary Approaches to Malaria Consortium. Artemisinin Resistance and the Unique Selection Pressure of a Short-acting Antimalarial. *Trends Parasitol* **36**, 884-887 (2020).
5. Hastings IM, Watkins WM, White NJ. The evolution of drug-resistant malaria: the role of drug elimination half-life. *Philos Trans R Soc Lond B Biol Sci* **357**, 505-519 (2002).
6. Alexander HK, MacLean RC. Stochastic bacterial population dynamics restrict the establishment of antibiotic resistance from single cells. *Proc Natl Acad Sci U S A* **117**, 19455-19464 (2020).